# Human spinal cord activation during filling and emptying of the bladder

Kofi A. Agyeman [1,2,3,4,5,10], Darrin J. Lee[1,3,4,5,10], Aidin Abedi[3,4], Sofia Sakellaridi[6], Evgeniy I. Kreydin[3,4,5,7], Jonathan Russin [1,3,4,5], Yu Tung Lo[3,4,8], Kevin Wu[3,4], Wooseong Choi[3,4], Sumant Iyer[3,4], V. Reggie Edgerton [4,5], Charles Y. Liu [1,3,4,5,6,7,11] ✉ & Vassilios N. Christopoulos[1,2,3,4,9,11] ✉

The spinal cord is essential for processing sensory information and regulating autonomic functions, such as bladder control, which is critical for urinary continence and voiding. Understanding how the spinal cord represents bladder pressure can provide valuable insights into the neural mechanisms underlying bladder control and contribute to developing better therapies for bladder dysfunction. However, measuring neural activity in the human spinal cord is notoriously challenging due to its small size and the surrounding bony and fascial enclosures, limiting the effectiveness of traditional neuroimaging techniques. Functional ultrasound imaging (fUSI) is a minimally invasive, emerging modality that overcomes these barriers, offering high sensitivity, spatial coverage, and spatiotemporal resolution for studying neural dynamics. Here, we combine fUSI with urodynamically controlled bladder filling and emptying to examine hemodynamic responses in the human spinal cord during one cycle of micturition. Using intravesical bladder pressure recordings, we identify spinal cord regions with hemodynamic signals that strongly correlate with bladder pressure. Furthermore, a linear support vector machine regression model (SVM-r) trained on the fUSI power Doppler signal reveals relevant spinal cord regions and accurately reconstructs bladder pressure changes. Our findings provide evidence of bladder pressure-responsive regions in the spinal cord, where hemodynamic signals strongly correlate with bladder pressure.

The spinal cord is frequently neglected in the study of neural function. As a result, its anatomy and physiology are not as well understood as those of the brain. Yet, it represents the first evolutionary step in central nervous system development and houses the neural circuitry that controls and modulates some of the most essential functions of life[1]. Neural networks capable of producing autonomous central commands – usually stereotyped and rhythmic motor behaviors – are present throughout the rostral and caudal parts of the spinal cord[2].

[1]Department of Biomedical Engineering, Viterbi School of Engineering, University of Southern California, Los Angeles, CA, USA. [2]Department of Bioengineering, University of California Riverside, Riverside, CA, USA. [3]Department of Neurological Surgery, Keck School of Medicine, University of Southern California, Los Angeles, CA, USA. [4]USC Neurorestoration Center, Keck School of Medicine, University of Southern California, Los Angeles, CA, USA. [5]Rancho Los Amigos National Rehabilitation Center, Downey, CA, USA. [6]Casa Colina Hospital and Centers for Healthcare, Pomona, CA 91767, USA. [7]Institute of Urology, Keck School of Medicine, University of Southern California, Los Angeles, CA, USA. [8]Department of Neurosurgery, National Neuroscience Institute, Singapore, Singapore. [9]Neuroscience Graduate Program, University of California Riverside, Riverside, CA, USA. [10]These authors contributed equally: Kofi A. Agyeman, Darrin J. Lee. [11]These authors jointly supervised this work: Charles Y. Liu, Vassilios N. Christopoulos. ✉e-mail: cliu@usc.edu; vchristo@usc.edu

Actions such as chewing, swallowing, and breathing are thought to be partially produced by these networks in the rostral cord[3]. Similarly, autonomic functions such as urination and defecation are under the control of neural networks located in the caudal spinal cord[4].

Although evidence for the existence of neural network circuits that control and regulate certain body processes is strong, its demonstration in humans has been challenging to achieve. The bony, fascial enclosure and small cross-section dimensions (approximately 12 mm in diameter) of the spinal cord, combined with susceptibility artifacts due to local magnetic field inhomogeneities generated by interfaces between surrounding bones, ligaments, soft tissues and cerebrospinal fluid (CSF), make the spinal cord an unfavorable target for traditional neuroimaging techniques, such as functional magnetic resonance imaging (fMRI)[5–11]. As a result, the bulk of our understanding of spinal cord function comes from animal and lesioning studies[12]. There is little direct evidence for function-specific spinal cord activity in humans, and fMRI, which has shed so much light on brain functions in humans, of the spinal cord is only minimally developed and generally restricted to the cervical cord[8–10,13]. Given this context, there is a clear and distinct need for developing neurotechnologies that make the functional study of the human spinal cord more accessible.

Functional ultrasound imaging (fUSI) is an emerging neuroimaging technology that represents a new platform with high sensitivity, spatial coverage and spatiotemporal resolution, enabling a range of new pre-clinical and clinical applications[14–23]. It was originally developed for brain imaging in small animals (i.e., rodents)[16]. Based on power Doppler imaging, fUSI measures changes in cerebral blood volume (CBV) by detecting backscattered echoes from red blood cells moving within its field of view[24,25]. While fUSI is a hemodynamic technique, its superior spatiotemporal performance (i.e., 100 μm and up to 10 ms) and sensitivity (~1 mm/s velocity of blood flow) offer substantially closer correlation to the underlying neuronal activities than other hemodynamic methods such as fMRI. It is minimally invasive and requires trephination in large organisms to enable penetration of ultrasound waves, as the skull attenuates acoustic waves. The fUSI scanner is akin to any clinical ultrasound machine, making it easily portable and negating the need for the extensive infrastructure inherent to fMRI.

Recently, fUSI was extended to study the spinal cord responses to electrical and mechanical stimulations in small animals and human patients[26–30]. Despite the significant contributions of these studies to understanding how the spinal cord responds to external sensory stimulation, none have demonstrated spinal cord circuits associated with physiological functions in humans. In the current study, we utilize fUSI to study the hemodynamic response of the spinal cord during urinary bladder filling and emptying in patients undergoing epidural spinal stimulation surgery for chronic low back pain treatment under general anesthesia. By combining fUSI recordings from the spinal cord with intravesical bladder pressure (BP) recordings, we identify spinal cord regions where the hemodynamic signal strongly correlates with bladder pressure. In addition, we reconstruct with high accuracy the pressure state of the bladder utilizing a linear support vector machine regression model (SVM-r) trained on fUSI spinal cord signals. Overall, our study provides an in-human application of fUSI to characterize the hemodynamic response of the spinal cord during urodynamically controlled bladder filling and emptying, opening avenues for better understanding of the mechanisms of control that the spinal cord exerts over micturition.

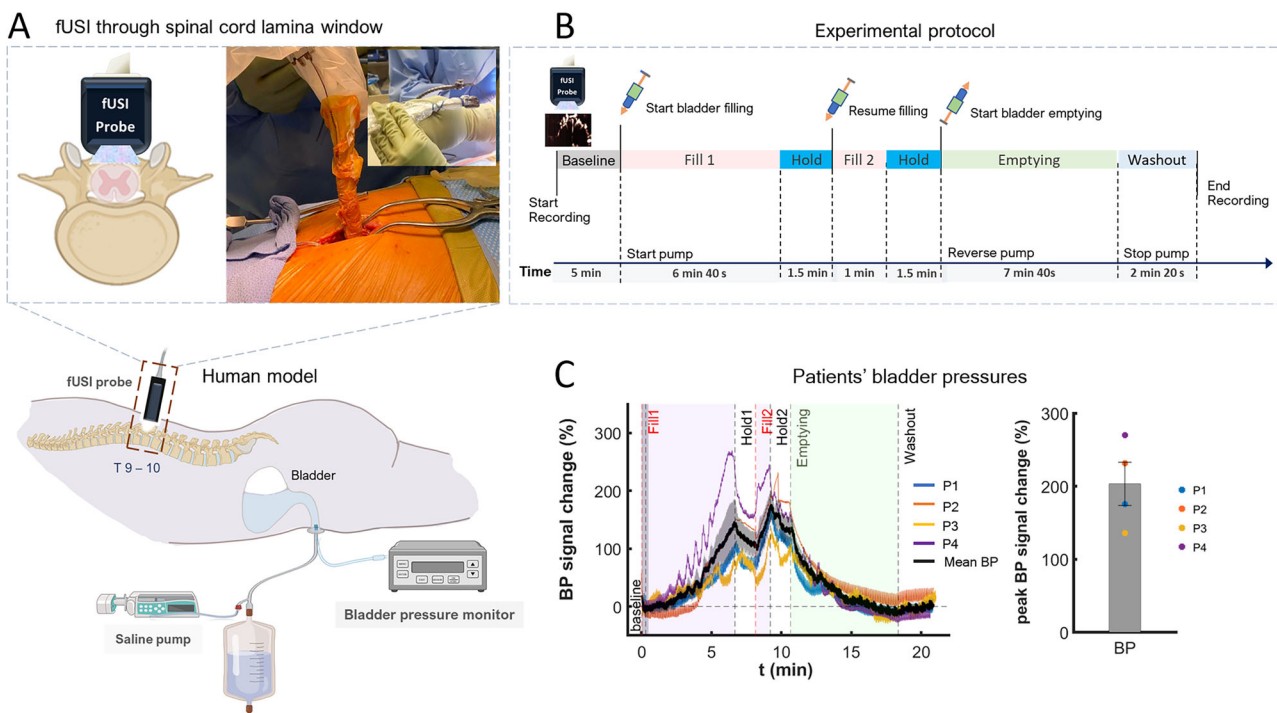

**Fig. 1 | Experimental setup and fUSI protocol. A** A graphical representation of the human urodynamic model developed to study spinal cord activity during micturition. The spinal cord fUSI acquisition was performed simultaneously during bladder filling and emptying through a laminar window using a miniaturized 15.6 MHz, 128-channel, linear ultrasound transducer array. Created in BioRender. A, K. (2025) https://BioRender.com/peg7anq and https://BioRender.com/oci3qlr. **B** The experimental protocol for urodynamically controlled filling and emptying of the bladder. **C** Bladder pressure (BP) signals percentage change over time during bladder filling and emptying for four patients (P1, P2, P3, P4 – blue, orange, yellow and purple, respectively) are shown in the left panel. The mean BP across all patients is represented by the black curve, with the standard error shaded in gray. The right panel shows a bar chart depicting the mean peak change (mean ± SEM) in bladder pressure across patients during filling and emptying, relative to the initial 30 s, with the standard error of mean bar in black. Individual peak changes for each patient are marked by colored dots (P1, P2, P3, P4 – blue, orange, yellow and purple, respectively). Source data are provided as a Source Data file.

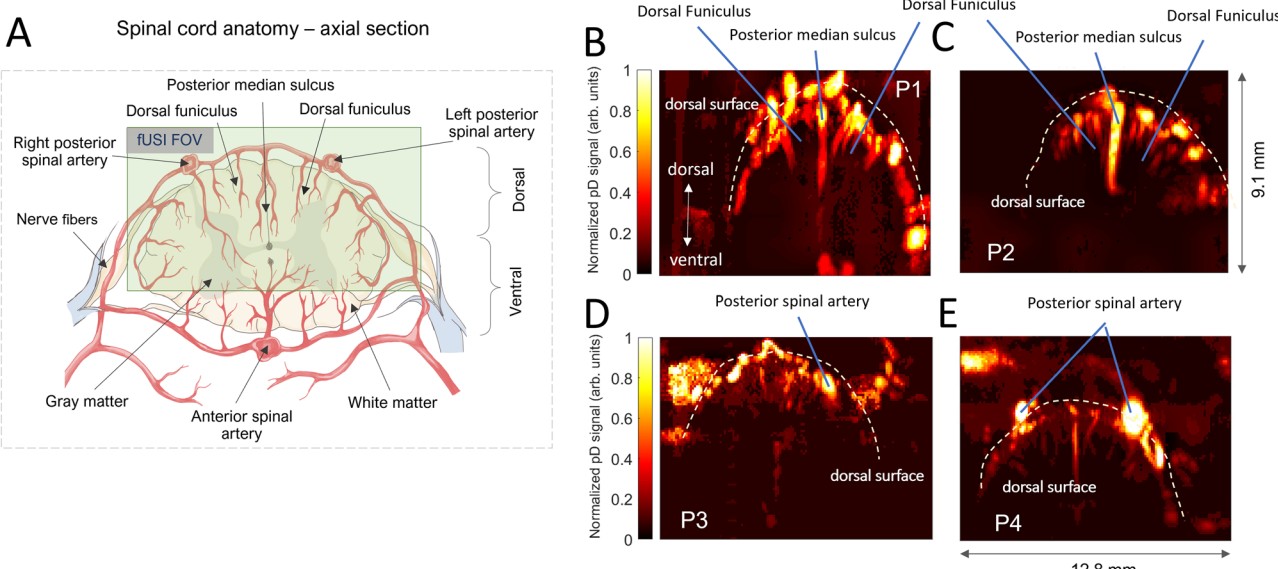

**Fig. 2 | Functional ultrasound imaging of the spinal cord in a transverse plane.**
**A** Cross section of spinal cord anatomy. The green area illustrates approximately the field of view of the fUSI acquisition is overlaid on a rendering of the spinal cord anatomy. **B**–**E** Power Doppler-based vascular maps showing the transverse section of the spinal cord of the four patients. The dorsal funiculus, posterior median sulcus and posterior spinal arteries are indicated on the spinal cord vascular maps, with the dorsal surface also designated by the white discontinuous lines.

## Results

To investigate how human spinal cord hemodynamics respond to bladder filling and emptying processes, we acquired fUSI images of the spinal cord from four (4) chronic low back pain patients, who underwent standard-of-care implantation of an epidural spinal cord stimulation (ESCS) device under general anesthesia (Fig. 1). The urodynamic experiment was performed before the ESCS implantation. A miniaturized 15.6 MHz, 128-channel, linear ultrasound transducer array was mounted on an articulating surgical arm fixed to the operating table and inserted through a partial laminar opening onto the dura at the level of the 10th thoracic vertebra (T10) with a transverse field of view (Fig. 1A). On average, an adult typically urinates every 3 to 4 h. However, we were limited to a total experiment duration of about 30 min, due to surgical time constraints. For this reason, we utilized a protocol that consisted of approximately 26 min of continuous fUSI signal acquisition, including 5 min of baseline activity, followed by a bladder filling cycle, interspersed with two hold periods (1.5 min each), an emptying cycle, and a washout period at the end (Fig. 1B). The hold period was implemented to simulate the natural micturition event in which individuals maintain continence despite a full bladder and to allow for the accommodation of rising bladder pressure as the bladder fills at a high rate (90 ml/min). The 1.5 min duration was selected based on previous experiments, which showed that this timeframe allows bladder pressure to equilibrate effectively[31]. The bladder was filled and emptied while continuous intravesical bladder pressure was recorded using a Laborie Goby™ (Vermont, USA) urodynamic system. The same protocol was employed for all patients. Figure 1C depicts the percentage changes of the bladder pressure (%BP) during filling and emptying relative to the bladder pressure signal acquired during the first 30 s, for all 4 patients (P1, P2, P3 and P4 in blue, orange, yellow and purple curves, respectively). The mean %BP signal change acquired across the 4 patients (black curve with the standard error indicated by the surrounding gray region) shows a gradual increase and decrease during the filling and emptying phases of the bladder, respectively. We observed a mean peak %BP of 203.23% ± 29.65% (Mean ± SE) across all patients (Fig. 1C, bar chart). The individual peak %BP for each patient is indicated by the colored dots (blue, orange, yellow and purple, patients P1, P2, P3, and P4, respectively). The non-normalized bladder

pressure curves in the recording units (cmH2O) across time for the four patients are presented in the Supplementary Section (Supplementary Fig. S2A). The variability in bladder pressure is expected, given the natural differences in bladder capacity and compliance among individuals – some people can accommodate larger fluid volumes than others. Note that only a single trial of bladder filling and emptying was performed for each patient, due to protocol constraints and time limitations during surgery.

### Hemodynamic response induced by bladder filling and emptying

Power Doppler (pD)-based functional ultrasound images were acquired from the spinal cord. The mean baseline pD signal of the spinal cord, measured during the 5 min preceding filling onset, captures the anatomical vascularization of the human spinal cord across all patients, with the dorsal surface indicated by the white discontinuous lines (Fig. 2B–E). The dorsal funiculus, posterior median sulcus and posterior spinal arteries are also indicated on the spinal cord vascular maps. The pD images have spatial resolutions of 100 μm × 100 μm in-plane, a plane thickness of about 400 μm, and a field of view (FOV) 12.8 mm × 9.1 mm. The FOV captures the dorsal and portions of the ventral cross-section of the spinal cord, approximately indicated by the light-green rectangular overlay on a spinal cord anatomical and vascular rendering in a transverse section (Fig. 2A).

To characterize the spinal cord hemodynamic response during filling and emptying of the bladder, we computed the spinal cord blood volume changes (ΔSCBV) – i.e., % pD signal changes – relative to the baseline activity (i.e., average pD activity measured for 5 min preceding the onset of bladder filling). The goal is to identify regions within the spinal cord where the hemodynamic signal correlates with bladder pressure. To do so, we computed the activation map for each patient by performing a Pearson's correlation analysis between the bladder pressure changes and ΔSCBV for each pixel in the spinal cord recorded area. The activation maps revealed spinal cord regions that are positively (reddish areas, correlation threshold: $r > 0.43$, z-score $> 0.49$) and negatively (blueish areas, correlation threshold: $r < -0.46$, z-score $< -0.53$) correlated with bladder pressure during filling and emptying of the bladder (Fig. 3A). Note that the correlation thresholds were chosen

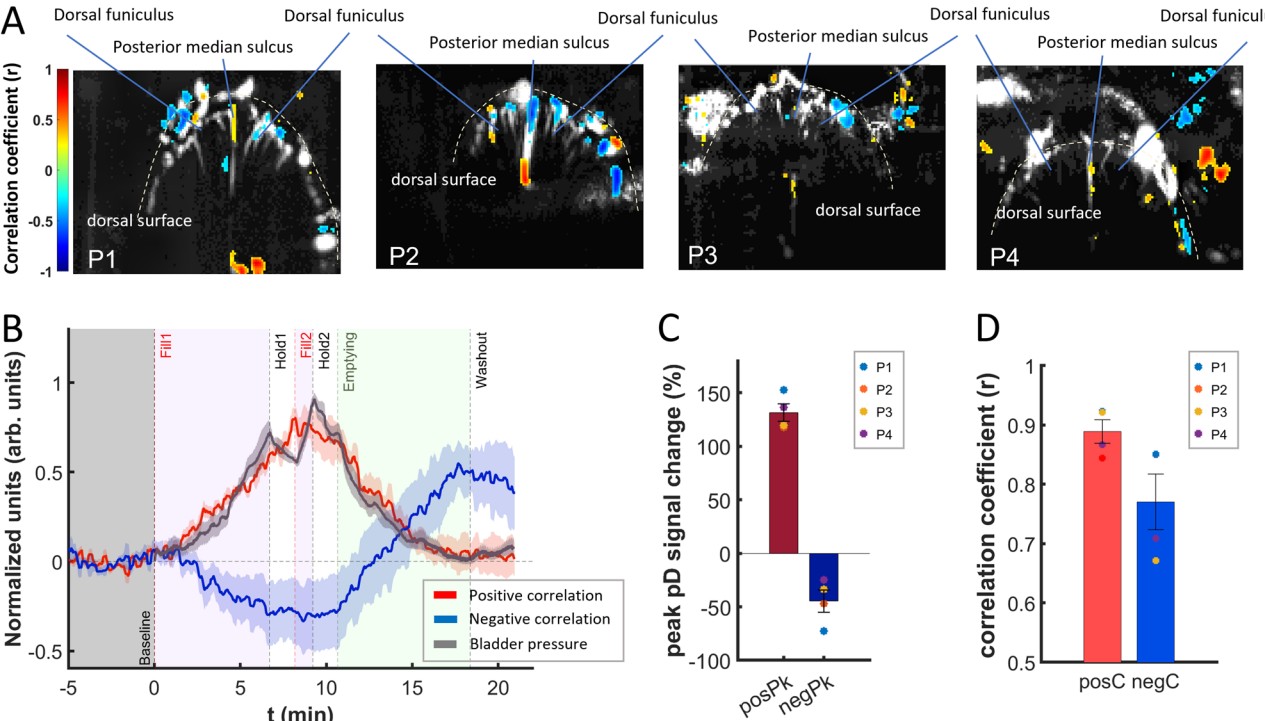

**Fig. 3 | Activation maps of the correlation between the spinal cord blood volume changes (ΔSCBV) and bladder pressure during filling and emptying of the bladder. A** Activation maps of the 4 patients illustrate spinal cord regions that are positively (reddish) and negatively (blueish) correlated with the bladder pressure during filling and emptying of the bladder. Spinal cord regions with significant positive (reddish areas, $r > 0.43$, z-score > 0.49) and negative (blueish areas, $r < -0.46$, z-score < −0.53) correlations to bladder pressure are depicted. The thresholds represent approximately the top 5% of significantly activated pixels. **B** Corresponding average ΔSCBV (normalized) of the spinal cord regions (*panel A*) that are positively (red curve) and negatively (blue curve) correlated with the bladder pressure across all patients. The gray curve depicts the normalized mean change of the bladder pressure during the urodynamic experiment across patients. The shaded regions around the bladder pressure and the ΔSCBV curves represent

the standard error derived from averaging across patients. **C** Average peak ΔSCBV (i.e., % pD signal change) from the baseline activity of the positive [red, posPk (positive peak)] and negative [blue, negPk (negative peak)] correlated bladder pressure-related spinal cord regions across all patients, with the standard error of mean bars (mean ± SEM) depicted in black. Individual peak ΔSCBV values during filling and emptying for each patient are indicated by colored dots (P1, P2, P3, P4 – blue, orange, yellow and purple, respectively). **D** Mean positive [red, posC (positive correlation)] and negative [blue, negC (negative correlation)] correlation coefficients (r) of the ΔSCBV of the activated spinal cord regions with bladder pressure across all patients, with the standard error of mean bars (mean ± SEM) depicted in black. Each of the 4 patients' positive and negative r-coefficients are indicated with colored dots (P1, P2, P3, P4 – blue, orange, yellow and purple, respectively). Source data are provided as a Source Data file.

to capture approximately the top 5% of both positive and negative correlations of spinal cord pixels to the bladder pressure. We selected these thresholds to avoid overcrowding the activation map and to better visualize areas where the pD signal is strongly correlated with the bladder pressure. Given the large number of voxel-wise comparisons, we applied false discovery rate (FDR) correction using the Benjamini-Hochberg (BH) procedure ($p < 0.01$) to control for false positives while maintaining statistical power. This approach is well-suited for neuroimaging studies with multiple statistical tests, ensuring that the reported correlations are robust and unlikely to result from chance.

To assess the temporal pattern of activation of the bladder pressure-related regions, we computed the average ΔSCBV over the top-ranked significant pixels of the positive and negative correlates to the bladder pressure, across time and patients, relative to baseline activity (5 min prior to bladder filling onset) (see Supplementary Fig. S2B). The results showed that the bladder manipulation caused strong neuroactivation in the spinal cord. We observed mean peak ΔSCBV responses of 123.33 ± 4.12 % (Mean ± SE) and − 42.61 ± 4.20 % for the top-ranked spinal cord positive and negative correlate-pixels across all patients (Fig. 3C). Given the variability in the magnitude of hemodynamic response between patients, we normalized the ΔSCBV values to a range of [−1, 1] for each patient. Similarly, we normalized the bladder pressure values to a range of [0, 1] to account for differences in the magnitude of pressure curves across patients. Figure 3B

shows the normalized ΔSCBV during filling and emptying the bladder for the positive (red curve) and negative (blue curve) correlate-pixels, along with the average normalized bladder pressure changes (gray curve) across all patients. The shaded regions around the bladder pressure and the ΔSCBV curves represent the standard error of the mean across patients. The average correlation between bladder pressure changes (i.e., %BP) and ΔSCBV is 0.89 ± 0.02 (Mean ± SE) for the positively (reddish) and − 0.78 ± 0.05 for the negatively (blueish) bladder pressure-related spinal cord regions (Fig. 3D).

Notably, we also observed bladder pressure-related regions extending beyond the dorsal surface, indicating that neural signals associated with bladder function may modulate hemodynamic activity in regions adjacent to the gray matter of the spinal cord. When we isolated and compared the mean ΔSCBV (i.e., %pD) signal averaged across activated regions outside the dorsal surface with that of activated regions within the spinal bounds, the results showed a consistent rise and fall of ΔSCBV during filling and emptying of the bladder across both activated spinal cord regions – i.e., within and outside the dorsal bounds (Supplementary Fig. S3A, B). Indeed, this suggests that the extra-spinal signal is related to the blood supply to the spinal cord itself. Intra-spinal perfusion is driven by extra-spinal arteries, and increased activation would be expected with elevated intra-spinal blood flow. Moreover, differences in activation between the left and right sides may influence these perfusion patterns. Notably, since the

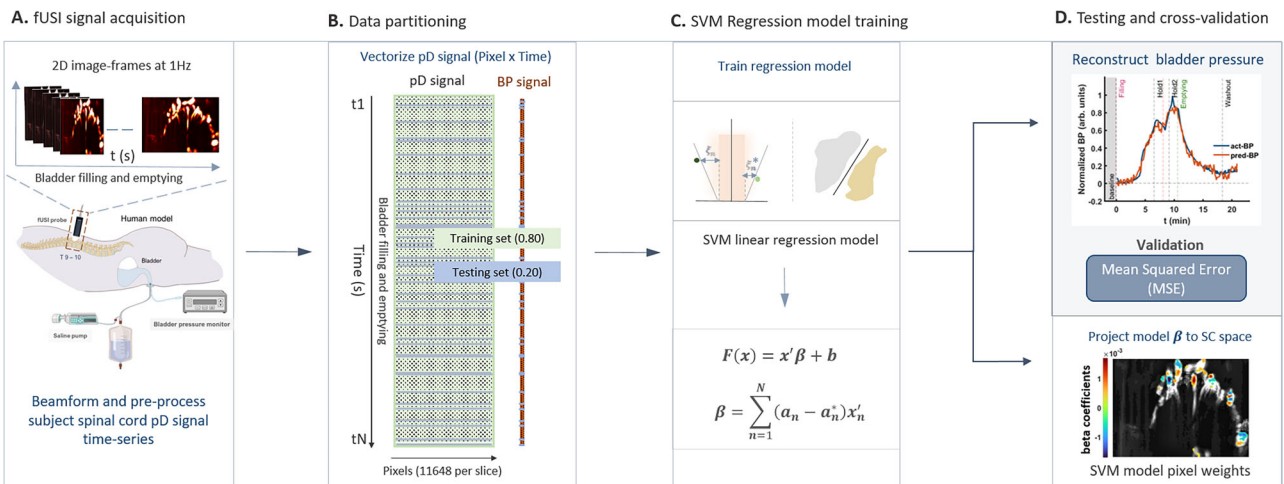

**Fig. 4 | Flowchart of the SVM-r algorithm developed to reconstruct bladder pressure. A** fUSI signal acquisition. fUSI data during filling and emptying of the bladder were recorded at the level of the T10 vertebral body and pre-processed to remove motion and non-experimental related artifacts. Created in BioRender. A, K. (2025) https://BioRender.com/peg7anq and https://BioRender.com/oci3qlr. **B** Data partitioning. fUSI and bladder pressure signals were vectorized and separated into training and testing images and bladder pressure values based on the cross-validation technique used – 80% training data and 20% testing data. Each dataset is composed of randomly selected image-frames and pressure values acquired during the entire process of filling and emptying of the bladder. Note that the corresponding time-point associated with the testing fUSI and bladder pressure data sets were stored to allow reconstruction of the predicted and actual bladder pressure curves during validation. **C** Support vector machine regression model training. A

regularized linear SVM-r algorithm was leveraged to train a regression model on the pD signal and to derive a prediction function that minimally deviates from the actual bladder pressure. **D** Testing and cross-validation. The performance of the SVM-r model was determined by evaluating the mean-squared-error between the actual bladder pressure and the corresponding reconstructed bladder pressure using the untrained pD spinal cord testing data. The SVM-r approach results in 1-dimensional beta ($\beta$) parameters that correspond to weighted coefficients of the predictor data (fUSI spinal cord image-frame pixels) variables. Mapping the $\beta$ coefficients onto the 2D spinal cord image space identifies important regions that encode the most relevant information for predicting the bladder pressure – the higher the $\beta$ coefficient, the more significant the contribution of this pixel to the bladder pressure dynamics.

activation occurs within the dura, it is unlikely to result from bleeding or extravasation. This further suggests that neural signals associated with bladder function may modulate hemodynamic activity in regions adjacent to the gray matter of the spinal cord. The activation detected in vessels outside the dorsal column is likely related to their role in supplying blood to the vasculature within the gray matter. Importantly, we have also observed spinal cord activation outside the gray matter in our recent epidural spinal cord stimulation study[30]. Finally, the extra-spinal signal is also unlikely to result from motion artifacts. If bladder manipulation (i.e., filling and emptying) induces significant motion in the spinal cord, we would expect all vascular pixels to exhibit a correlation with bladder pressure, as such motion would likely result in a uniform, rigid body shift across the pD image. Note that we account for and remove motion artifacts from the pD images to ensure that observed hemodynamic changes reflect true neurovascular activity rather than movement-related artifacts (Supplementary Fig. S3C and D, and Supplementary Movies 1 and 2).

**Reconstructing bladder pressure from the spinal cord pD signal**
The ability to reconstruct the temporal dynamics of bladder pressure from the pD signal would be a significant step toward developing assistive spinal cord interface systems for bladder function restoration. The current experimental design was limited to a single micturition cycle, potentially with inter-frame dependencies in the pD images, which restricts our ability to generalize the model for predicting future bladder pressure changes. To enable inductive learning approaches capable of such generalization, multiple micturition cycles would be required. However, incorporating multiple cycles would extend the experiment beyond the time constraints imposed by the clinical protocol. Consequently, we employed a transductive learning approach, focusing on reconstructing bladder pressure from the current pD signal alone. To do so, we leveraged a non-parametric SVM linear regression (SVM-r) algorithm[32,33] (Fig. 4). After removing non-experimental related

fluctuations from the pD recordings (Supplementary Fig. S1 and Supplementary Movies 1 and 2), the entire (baseline to washout) 2D spinal cord fUSI image frames acquired from each patient were aligned in time (Fig. 4A). To train an SVM-r model, each patient's spinal cord %pD 2D time series, acquired during filling and emptying of the bladder was vectorized into 1D vectors and aligned in rows to form a 2D predictor data matrix (pixels × time). This was paired with the corresponding changes in the bladder pressure signal (%BP) relative to the first 30 s of the bladder pressure recording to form a response data vector (Fig. 4B). The pD data matrix (predictor) and the BP (response vector) were randomly partitioned into training (1004 time points, 80%) and testing (250 time points, 20%) subsets for cross-validation analysis. Note that the training and testing datasets are composed of randomly selected image frames and pressure values acquired during the entire urodynamic experiment. The corresponding time-instants associated with the pD and BP testing datasets were stored to allow reconstruction of the predicted and actual bladder pressure curves. Next, we leveraged a regularized SVM-r algorithm to train a regression model on the pD signal that determines a function that minimally deviates from the bladder pressure. The algorithm minimizes an objective function to optimally reconstruct (i.e., predict) the bladder pressure (Fig. 4C) (see Materials and Methods section for details). The analysis was performed separately for each patient ($N = 4$). We adopted the linear SVM-r approach, which is well-suited for high-dimensional sparse data like ours[34–36]. It ensures a computationally simple solution for the optimization problem by utilizing a predictor function that depends solely on support vectors with beta ($\beta$) parameters that can be described completely as a linear combination of training observations[37].

The cross-validation analysis demonstrated the capability of the SVM-r model to reconstruct bladder pressure changes during filling and emptying of the bladder with high accuracy (Fig. 4D and Supplementary Fig. S4, patient 1). Overall, the model accurately reconstructed the temporal dynamics of bladder pressure during the micturition

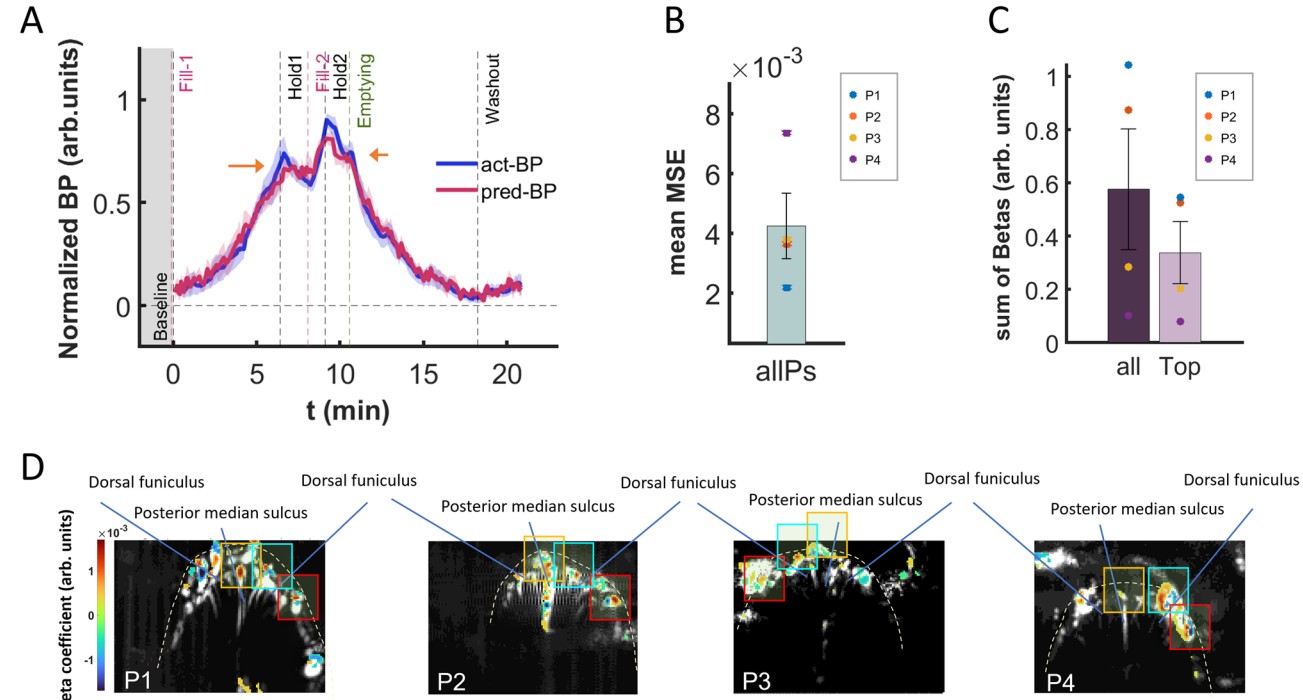

**Fig. 5 | Bladder pressure reconstruction and identification of related spinal cord regions using SVM-r. A** Actual (blue curve) and reconstructed (red curve) average bladder pressure across all patients. The SVM-r reconstructed bladder pressure (pred-BP), determined using untrained validation fUSI spinal cord testing data, closely captures the trends of the actual bladder pressure (act-BP). The shaded regions around the act-BP and pred-BP curves represent the standard error derived from averaging across patients. **B** SVM-r prediction performance. The average mean-squared-error between the actual and reconstructed bladder pressure across all patients – with the standard error of the mean bar (mean ± SEM) depicted in black. Each patient's mean squared error (MSE) is indicated by the colored dots (P1, P2, P3, P4 – blue, orange, yellow and purple, respectively). **C** Average sum-total of all and the top 5% SVM-r algorithm $\beta$ coefficients used for visualizing the bladder pressure related spinal cord regions across patients – with the standard error of mean bars (mean ± SEM) depicted in black. The $\beta$ coefficients represent weights derived from the SVM-r model training and assigned to each predictor variable (fUSI spinal cord image pixel). Each patient's sum-total of all and the top 5% $\beta$ coefficients are indicated by the colored dots (P1, P2, P3, P4 – blue, orange, yellow and purple, respectively). **D** Weighted spinal cord region maps of patients P1 to P4 derived from the linear SVM-r algorithm using fUSI recorded data. The top 5% SVM-r $\beta$ coefficients are displayed as color-coded overlay on the mean grayscale pD spinal cord vascular map derived during baseline, with reddish and bluish regions indicating positive and negative $\beta$ coefficients, respectively. These regions represent important spinal cord areas that encode the most relevant information for predicting bladder pressure. Source data are provided as a Source Data file.

cycle, including also the pressure drop in the hold periods (red curve: predicted [i.e., reconstructed] mean bladder pressure across all patients), closely matching the actual recorded bladder pressure signal (blue curve: actual mean bladder pressure signal) (Fig. 5A). The shaded regions around the bladder pressure curves represent the standard errors resulting from averaging the bladder pressure across patients. The high reconstruction accuracy was quantified and reflected by a low mean-squared-error (MSE) [0.0042 ± 0.0011 (Mean ± SE)] average attained across patients (Fig. 5B). For the actual and reconstructed bladder pressure for each of the 4 patients, see Supplementary Fig. S4.

Finally, we projected the SVM-r $\beta$ coefficients back onto the 2D spinal cord image space for each patient to identify spinal cord regions where the ΔSCBV signal reconstructs bladder pressure. These coefficients represent the weights assigned to each predictor variable, indicating the relative importance of each variable (i.e., pD spinal cord image pixel) in reconstructing bladder pressure during the regression model training process. Figure 5C shows the averaged sum of $\beta$ coefficients (0.58 ± 0.23, sum-total [Mean ± SE]) alongside the average of the sum of the top 5% highest $\beta$ coefficients (0.34 ± 0.12, sum-total). In addition, Fig. 5D displays the top 5% $\beta$ coefficients as color-coded overlay regions on a gray-scale background of the mean baseline spinal cord pD signal for each patient. We identified consistent spinal cord regions weighted for bladder filling and emptying across all patients. Two of the prominent weighted regions are bifurcated by the dorsal funiculus (cyan and red boxes), while the third region appears

consistently over the posterior median sulcus of the spinal cord (orange box) in all patients. Although we observe similar weighted spinal cord pixel-regions, in contrast to all other patients, the boxed regions in patient 3 (P3) are stronger in the left hemisphere. In addition, compared to the activation maps result from the correlation analysis, it is worth noting that most of the top-ranked $\beta$ coefficients are contained within the spinal cord dorsal surface (white discontinuous lines). The linear SVM-r analysis extends beyond conventional statistical correlations by considering the entire pD image space and incorporating additional factors, such as spatiotemporal relationships between pixels. This allows the model to be trained in a more comprehensive manner, identifying relevant spinal cord regions that may explain the observed differences between the weighted (i.e., SVM-r) and activated (i.e., pixelwise correlation analysis) pixel regions. For similar effects observed between the activation maps generated by conventional and machine learning analyses, refer to Agyeman et al. on spinal cord stimulation[30].

## Discussion
### General

While functional neuroimaging in the human brain has advanced our understanding of brain function in micturition[31,38,39], the neural mechanism in the human spinal cord that controls filling and emptying of the bladder is largely unknown. To the best of our knowledge, there is no study that has characterized hemodynamic changes in the human

spinal cord during filling and emptying of the bladder. This lack of research is likely due to the complex structure of the spinal cord, including its small cross-sectional area, the cardiac-related motion of cerebrospinal fluid (CSF), and motion artifacts caused by the proximity of organs such as the lungs. These factors make the spinal cord an unfavorable area for conventional functional neuroimaging studies[5,8]. Electrophysiology suffers from the inherent trade-offs between sampling density, coverage and channel count, making it challenging to achieve a spatial sampling resolution of less than 100 μm over a large, recorded volume (i.e., 1 cm$^3$ would require about 10$^6$ channels). In addition, electrophysiological recordings in the human spinal cord are unlikely due to the risks associated with penetrating the spinal cord and causing trauma. Optical imaging is capable of monitoring single-neuron activity over large areas, but is typically limited by a penetration depth of < 1 mm[40,41].

Within this context, fUSI represents an emerging neuroimaging technology that utilizes ultrasound to monitor blood flow changes as an indirect readout of neuronal activity with high spatiotemporal resolution, penetration depth and sensitivity to slow blood flow motion. Originally developed for brain neuroimaging, fUSI has been recently expanded to study spinal neurovascular responses in small animals[26–29] and human patients[30]. Although these studies provide significant insights into better understanding spinal cord physiology in sensory integration, they are limited to artificial external stimulation and the prediction of discrete spinal cord states – i.e., stimulation on vs. stimulation off. In the current study, we took the next major leap in fUSI spinal cord research by recording functional activity of the human spinal cord during urodynamically controlled bladder filling and emptying. Here, we show that fUSI can detect hemodynamic signals in spinal cord regions that are highly correlated with the bladder pressure. We also introduce a machine learning algorithm that can identify important spinal cord regions that can accurately reconstruct bladder pressure changes using the pD signal from the spinal cord. Overall, the ability to characterize and correlate spinal cord hemodynamics with urodynamically controlled micturition events offers significant potential for further understanding the functional and dysfunctional anatomy associated with lower urinary tract physiology.

### Neuroscience and scientific applications

The unique combination of fUSI technology with anatomically correlated and easily monitored physiological function of micturition – mimicked by urodynamically controlled filling and emptying of the bladder – opens new opportunities for better understanding of the spinal cord networks that promote urinary storage and induce urinary emptying. It also creates avenues for studying the neural circuitries that control and modulate other important bodily functions, such as sensation, ambulation (e.g., passive movements in anesthetized patients). The detection of both positively and negatively correlated spinal cord regions with the bladder pressure during one cycle of micturition suggests the involvement of both excitatory and inhibitory spinal cord networks in the micturition process. These findings are consistent with the literature showing that spinal cord regions in the thoracic, lumbar, and sacral segments regulate urinary storage and voiding[42,43]. During bladder filling, sympathetic pathways primarily provide inhibitory control to prevent premature voiding[12]. For urination, this inhibition decreases, enabling bladder contraction and urethral sphincter relaxation through coordinated excitatory and inhibitory interactions essential for normal micturition.

In addition, the existence of spinal cord regions, where hemodynamic signal reconstructs the temporal dynamics of the bladder pressure changes, provides the proof-of-concept for developing ultrasound-based spinal cord machine interface technologies for bladder control in patients with neurogenic lower urinary track dysfunction. Surveys have repeatedly revealed that restoration of bladder function remains the top priority for spinal cord injury patients, far

ahead of even restoring the ability to walk[44]. Beyond spinal cord injuries, an even larger population worldwide suffers from urinary dysfunctions of neurological origin, with other etiologies. Developing spinal cord machine interfaces for informing patients about the state of the bladder would be a step closer to restoring bladder control.

### New avenues for improving neuromodulation treatments for neurogenic lower urinary tract dysfunction

Urinary dysfunctions of neurological origin due to spinal cord or brain injury, degeneration, or stroke represent some of the biggest medical burdens in the world and lead to uniquely dehumanizing consequences[45]. Therapies that are currently available abate some symptoms of neurogenic lower urinary tract dysfunction, but none can restore normal function. On the other hand, neuromodulatory approaches such as epidural spinal cord stimulation (ESCS) of the lumbosacral spinal cord have shown potential to activate neural networks associated with bladder function in rodents with spinal cord injury and thus lead to a degree of functional recovery[46,47]. In addition, clinical studies have shown that transcutaneous electrical spinal cord stimulation (TSCS) – i.e., a non-invasive neuromodulation therapy that stimulates the spinal cord from the surface of the skin – can reengage the spinal circuits involved in bladder control and normalize bladder and urethral sphincter function in patients with spinal cord injury[48,49]. Although neuromodulation therapies offer great promises for restoring normal lower tract function, their mechanism of action (MOA) remains elusive. This is mainly due to the lack of a monitoring modality that can characterize the effects of neuromodulation on spinal cord activity. Combining fUSI with neuromodulation of spinal networks has considerable potential in gaining a better understanding of the MOA of neuromodulation and augmenting its efficacy in improving bladder control in patients with neurogenic lower urinary tract dysfunction. Fine-tuning stimulation wave properties, such as amplitude, frequency, and shape, using fUSI has the potential to facilitate the objective identification of efficacious targets for neuromodulation.

### Limitations and clinical challenges

While fUSI is a novel technology that enables monitoring of brain and spinal cord activity, the skull and the lamina bones attenuate and result in aberrant acoustic waves at high frequencies, substantially reducing signal sensitivity. For this reason, most fUSI applications are minimally invasive, with few exceptions such as in young mice (8–12 weeks old with thin skull)[50] and in pediatric transfontanelle-imaging[17,51]. Surgical procedures to produce a craniotomy[21] or thinned-skull window[52] in brain research and laminectomy[26,30] in spinal cord research are required to harness the host of fUSI benefits. Monitoring non-invasively the spinal cord neuroactivation with fUSI in awake adults is challenging and has yet to be proven. However, recent studies in brain research provide evidence that non-invasive fUSI is capable either through a permanent "acoustic window" installed as part of a skull replacement procedure following a decompressive hemicraniectomy (partial skull removal)[53] or by intravenously injecting microbubble contrast agents for enhancing the fUSI signal[54,55]. Although these approaches have not yet been tested in spinal cord research, the promise of fully noninvasive fUSI in the spinal cord is imminent.

To overcome the limitations of invasive fUSI, one could argue that non-invasive traditional ultrasound methods may be used to estimate bladder size and send notifications to mobile devices regarding bladder status. However, the purpose of our study is to provide deeper insights into how bladder pressure correlates with spinal cord activity. By characterizing the hemodynamic response in the spinal cord during urodynamically controlled micturition, we can identify regions that predict bladder pressure, which is crucial for understanding the spinal mechanisms of normal bladder control and identifying abnormalities in urinary incontinence patients. Our study provides a basis for exploring micturition mechanisms in the human spinal cord, which is

crucial for developing effective interventions. Furthermore, revealing neuroactivation patterns during micturition may optimize neuromodulatory therapies, such as TSCS, and contribute to developing spinal cord-machine interfaces for bladder restoration, allowing patients to regain the sensation of urgency to void.

In addition, it is important to acknowledge that we recorded activity in the thoracic cord (T10 lamina), although the main control mechanism of the bladder is thought to be located in the sacral cord between S2–S4, with the major portion at S3[2,56]. This is a common limitation in clinical studies as we often cannot record activity directly from the most desirable locations. In our study, we image the spinal cord during urodynamically controlled micturition in patients undergoing ESCS surgery for chronic low back pain treatment. The midline of the spinal cord at the T10 lamina is the preferred location for insertion of a more rostral spinal cord stimulator, and therefore, the laminectomy allows us to perform functional neuroimaging only at the T10 region. However, this clinical limitation does not affect our main finding that the hemodynamic signal within the T10 vertebral body correlates with the bladder pressure. In fact, this finding supports the prevailing hypothesis that micturition is regulated by neural circuits that traverse the entire central nervous system from the sacral plexus to the prefrontal cortex and vice versa. When the sacral plexus receives the sensory information from the bladder, this signal travels up the spinal cord to higher centers in the pons and above[12]. Also, the signal from the brain in turn travels down to the spinal cord to make sure that we only urinate when and where it is appropriate[57]. Therefore, it is likely that the bladder pressure-related signal that we detect at the T10 vertebral body level is a combination of the signal initiated in the sacral plexus that traveled towards higher brain centers, and the signal that is transferred from the brain to the bladder through the spinal cord. It is worth noting that our finding is consistent with a recent study on TSCS, showing that stimulation at the lower thoracic level can augment LUT function in non-human primates within a single session[58]. This implies the existence of a signal associated with micturition in the lower thoracic region.

Furthermore, while it is common in animal spinal cord studies to perform large laminectomies, retract back muscles and remove connective tissues[26–29], the surgical protocol cannot be modified to improve the quality of the fUSI images in human experiments. Instead, we performed only partial and small laminectomies to avoid spine destabilization. In particular, the width of the laminar opening (about 11 mm) was often smaller than the width of the ultrasound probe (12.8 mm) and consequently the probe did not abut perfectly to the dura. Therefore, it is challenging to image the same 2D plane across patients. Although the imaging planes vary slightly across the 4 patients, this does not affect the spatiotemporal pattern of the hemodynamic signal in the bladder pressure-related regions. Note that the variability observed in the peak of the %pD signal changes during filling and emptying of the bladder is not necessarily due to differences in the imaging plane across patients. It could also be attributed to other factors such as variations in bladder capacity and compliance, i.e., some individuals need to void more frequently than others. Nevertheless, the ability of fUSI to identify bladder pressure-responsive regions across patients highlights the strength and robustness of this neuroimaging technology to overcome the potential to image different 2D slices of the spinal cord across patients.

We also need to point out that the clinical protocol prevented us from using clamps to stabilize the spinal cord, a method commonly employed in animal neurophysiological studies to minimize cardiac and respiratory motion artifacts[59–61]. This raised concerns that the pD images might be affected by motion artifacts. However, we effectively mitigated these artifacts by securely mounting the fUSI probe on an articulating surgical arm attached to the operating room table and applying advanced pre-processing techniques previously validated in

human brain[21,53] and spinal cord fUSI studies[30] (see Supplementary Figs. S1 and S3C, D and Supplementary Movies 1 and 2).

Furthermore, the current study was limited to a single micturition cycle due to the time constraints of the surgical procedure. This restricts our ability to assess whether the SVM-r model, which was trained on a subset of data, potentially with inter-frame dependencies between pD images, from a single micturition cycle, can generalize to predict bladder pressure changes in future pD recordings. Despite this limitation, the transductive learning approach used in this study is significant, as it successfully identified spinal cord regions where pD signals can reconstruct bladder pressure during filling and emptying. Future animal studies incorporating multiple micturition cycles could help evaluate the generalizability of the pD-based SVM-r model for predicting bladder pressure changes.

In addition, we acknowledge that anesthesia can influence blood flow and, consequently, the pD signal. However, we do not expect a major impact on our findings for two reasons. First, the level of anesthesia was maintained roughly constant throughout the urodynamic experiment. Second, we recorded baseline activity prior to bladder filling and emptying, allowing the %pD signal change to be calculated relative to a stable baseline. This approach effectively mitigates the confounding effects of anesthesia, as it remains constant across the experiment, ensuring that observed changes in the pD signal reflect bladder activity rather than anesthetic influence. Nevertheless, it would be valuable to extend the fUSI-urodynamic experiment to awake humans once the ongoing development of fUSI technology allows for non-invasive imaging. Moreover, the patients included in the current study have chronic back pain, but do not present with any urinary system dysfunction. They are on medication solely for pain management and have no history or symptoms of urinary incontinence or other urological conditions. Therefore, we do not expect their chronic pain conditions to affect the findings related to micturition, as their urinary systems function normally. Finally, this study was performed using the Iconeus One scanner, where the fUSI pulse sequence was optimized for rodent brain imaging rather than for the human spinal cord. Despite this, the results demonstrated that fUSI technology is highly robust, enabling the identification of bladder pressure-responsive regions in the spinal cord, even with a non-optimized pulse sequence. In future studies, the pulse sequence can be further optimized specifically for human spinal cord imaging to enhance the accuracy of the findings.

Taken together, we present an in-human characterization of spinal cord hemodynamics during physiologically driven bladder activation. By combining fUSI with urodynamically controlled bladder filling and emptying, we identify regions whose activity correlates with bladder pressure, providing evidence for functional networks involved in micturition. These findings open new avenues for investigating spinal cord circuits that regulate not only bladder control but also other physiological processes in both health and disease.

## Methods
### Patient and surgical procedures
Four participants (2 females and 2 males) were included in the current study. Sex and/or gender was not a variable of interest in the study design and was determined based on self-report. Analyses of potential sex- or gender-related differences were not conducted, as they were beyond the scope of this study. The participants were recruited from patients who underwent standard-of-care implantation of a spinal cord stimulator paddle lead (PentaTM model 3228) at the Keck School of Medicine of the University of Southern California (USC). All patients were diagnosed with failed back surgery syndrome, which required a T10 partial laminectomy for insertion of a stimulation paddle lead in the prone position under general anesthesia. The fUSI probe was securely mounted on an articulating surgical arm attached to the operating room table to ensure stability throughout the imaging process and positioned

over the spinal cord at the T10 vertebral body level in a transverse orientation to acquire pD images during bladder filling and emptying. Notably, the urodynamic experiment was performed before the paddle lead placement (Fig. 1A). Informed consent was obtained from all patients after the nature of the study and possible risks were clearly explained, in compliance with protocols and experimental procedures approved by the USC Institutional Review Board.

## Patient bladder pressure signal acquisition

The urodynamic assessments in this study were conducted using the Laborie Goby (TM) urodynamics system for filling, emptying, and acquiring continuous intravesical bladder pressure measurements of patients. A LaborieT-DOC (TM) catheter was inserted into the bladder after patients were anesthetized. The position was confirmed by irrigation and aspiration. The infusion port of the catheter was connected to a drainage bag, and the manometer port was connected to the Laborie UDS Roam Bluetooth transmitter. The patients were then positioned prone. To begin the experiment, the infusion port of the catheter was connected to the infusion tubing and fUSI recordings were performed simultaneously with the urodynamics (See details of the experimental protocol below).

## Functional ultrasound imaging data acquisition

The spinal cord hemodynamic signals were acquired with a fully featured commercial Iconeus One scanner (Iconeus, Paris, France). A 128-element linear array transducer probe (IcoPrime, Iconeus, Paris, France) with a 15.6 MHz center frequency (60% bandwidth) and 0.1 mm pitch was inserted through the laminar opening to generate fUSI images (Fig. 1A). This approach enabled image acquisition with spatial resolution of 100 μm × 100 μm in-plane, slice thickness of 400 μm, and FOV of 12.8 mm (width) × up to 10 (depth) mm. The penetration depth was sufficient to image the dorsal portion and part of the ventral portion of the spinal cord on a transverse orientation. The probe was fixed steadily on a surgical articulating arm attached to the operating room table throughout experiments with the FOV transverse and intersecting the spinal cord central canal (Fig. 2A). Each image was obtained from 200 compounded frames using 11 tilted plane waves separated by 2° (i.e., from − 10° to + 10° increment by 2°), at a 500 Hz frame rate. Imaging sessions were performed using a real-time continuous acquisition of successive blocks of 400 ms (with 600 ms pause between pulses) of compounded plane wave images, with a 5500 Hz pulse repetition frequency (PRF). The acoustic amplitudes and intensities of the fUSI sequence remained below the Food and Drug Administration limits for ultrasonic diagnostic imaging (FDA, 510 (k)), yielding a mechanical index of 0.44, intensity spatial peak temporal average (ISPTA) of 120 mw/cm$^2$, and a maximum temperature rise + 1.2 °C at the lens contact.

## Experimental protocol

A 26 min continuous fUSI signal acquisition protocol was employed for all patients. The protocol consisted of 5 min baseline fUSI recording of the spinal cord, followed by a urodynamic experiment with simultaneous acquisition of intravesical bladder pressure and fUSI signal. This included a bladder filling cycle with two 1.5 min hold periods, an emptying cycle, and a washout period at the end (Fig. 1B). At the 5 min mark, Switching to passive tone: "the bladder was filled through a catheter with 600 ml of saline at a rate of 90 ml/min for 6 min and 40 s, while simultaneously recording the bladder pressure. The filling was paused for about 1.5 min, followed by additional bladder filling with saline for about 1 min. We then stopped the pump for 1.5 min and reversed the pump to continuously withdraw saline via the catheter for 7 min and 40 s at a rate of 90 ml/min, with continuous recording of the bladder pressure. The pump was turned off, then followed by 2 min and 20 s of additional fUSI spinal cord and bladder pressure signal recordings.

## Data preprocessing

Built-in phase-correlation-based sub-pixel motion registration[62] and singular value decomposition (SVD)-based clutter filtering algorithms[63], implemented by the Iconeus One scanner, were used to separate tissue motion signal from blood signal to generate relative pD signal intensity images[64]. We adopted rigid motion correction techniques[65] that have been successfully used in fUSI[21,23,30] and other neuroimaging studies[66–68], to address potential physiological and motion artifacts unique to human spinal cord imaging. This was combined with in-house high-frequency smoothing filtering. We utilized a lowpass filter with normalized passband frequency of 0.04 Hz, with a stopband attenuation of 60 dB that compensates for delay introduced by the filter, to remove high-frequency fluctuations in the pD signals · high frequency fluctuations are often associated with noise and artifacts, such as motion, vibration from the scanner (i.e., electronics, power supply, probe, etc) and other non-physiological fluctuations. The application of the filter assumes that fluctuations in the pD signal above a frequency threshold are likely artifacts and not related to the experimental effects. The reason is that fUSI detects hemodynamic changes that are known to be relatively slow. Further, the acquisition parameters employed in this study resulted in a final image acquisition frame rate of 1 Hz (i.e., a final spinal cord pD vascular map image was acquired every 1 s). Supplementary Fig. S1 shows the unfiltered and filtered mean pD vascular map of the spinal cord acquired during baseline, and the pD signal changes of selected ROIs, before and after pre-processing, respectively, in a typical patient (P1). In addition, the Supplementary Section presents movies of the sequence of pD images before and after motion correction (Supplementary Movies 1 and 2, patient P4). In the unfiltered movie (Supplementary Movies 1, sped up to 20 frames per second), motion artifacts are evident, highlighted by the blue arrow. In contrast, after applying motion correction, the pD signal is stabilized, and motion fluctuations are eliminated in the filtered movie (Supplementary Movie 2, also sped up to 20 frames per second). Both movies depict the pD spinal cord activity recorded during the 5 min baseline period and the subsequent first-filling phase, totaling 11 min and 40 s, from patient P4. Following signal preprocessing, we observed a reduction in the mean standard deviation of the pD signals across all pixels over the duration of the recordings (i.e., movies) for patient P4, from 10.88 ± 0.43 arb. units (Mean ± SE) in the raw data to 6.34 ± 0.25 arb. units in the filtered data.

## Spatiotemporal correlation of bladder pressure changes to ΔSCBV

We assessed the spatiotemporal effects of bladder filling and emptying on spinal cord hemodynamics. We generated pixel-wise activation time course curves of ΔSCBV as a percentage change of the pD signal relative to baseline activity for the whole spinal cord FOV. The mean pD signal activity acquired 5 min preceding the onset of the bladder filling was utilized as the baseline for the analysis. We investigated whether there are spinal cord regions where ΔSCBV is correlated with the bladder pressure during filling and emptying. To test this hypothesis, we computed Pearson correlation coefficients for each pixel in the spinal cord fUSI image. To this end, the %pD signal from each pixel is correlated with the bladder pressure change (%BP) across time for each participant to determine pixels with statistically significant correlation ($p < 0.01$, with Benjamini-Hochberg procedure to control for false discovery rate [FDR] correction[69]). We generated activation maps of the pixels that show significant positive and negative correlations above an r-coefficient threshold ($r > 0.43$, z-score > 0.49 and $r < − 0.46$, z-score < − 0.53) respectively. Note that the r-coefficient threshold corresponds to the top-ranked spinal cord pixels with the top 5% r-coefficients. Finally, to visualize the temporal dynamics of the %ΔSCVB, we derived mean %pD signal change curves by averaging the signals from the top 5% of pixels showing significant correlation with the bladder pressure signal.

## Reconstructing bladder pressure dynamics from SCBV signals

Next, we leveraged an unbiased and non-parametric transductive machine learning approach to reconstruct the bladder pressure temporal dynamics and to identify spinal cord regions with a ΔSCBV signal that best captures the temporal changes of the bladder pressure. To do so, we employed a SVM-r learning algorithm[32,33,70,71] that includes the following steps: (1) time-align the spinal cord pD signals (i.e., images) acquired during filling and emptying of the bladder, (2) vectorize the 3D pD signal ($z \times y \times t$ – space × time) into 2D ($p \times t$ – pixels × time) space, and align with corresponding 1D bladder pressure ($b \times 1$) vector (i.e., BP signal), (3) train a linear regression model utilizing a regularized SVM-r method that optimally predicts the bladder pressure, (4) cross validate, evaluate the decoder performance and identify relevant spinal cord areas where the pD signal predicts bladder pressure dynamics (Fig. 4). To this end, we determined the percentage change in pD signal in each pixel in the spinal cord fUSI images (fillings, holds and emptying periods) for each patient, relative to the baseline signal activity. The signal acquired 5 min before the onset of filling was used to calculate the %pD signal change for each pixel. The entire pD spinal cord 2D image space was utilized in the machine learning algorithm (Fig. 4A). Each 2D time series dataset was vectorized into 1D vectors and aligned in rows to form the 2D predictor data matrix (pixels × time). To form a response data vector (BP signal), each patient's corresponding change in the bladder pressure signal relative to the first 30 s of the bladder pressure was aligned with the predictor matrix (Fig. 4B). Next, we leveraged a regularized linear SVM-r algorithm to train a regression model on the pD signal that optimally determines a function (Eq. 1)

$$F(x) = x'\beta + b, \tag{1}$$

that minimally deviates from the bladder pressure, while remaining as flat as possible (where $x'$ is an observation from a set of predictor variables, i.e., time step during filling and emptying with the pD image pixels as the predictor variables; $\beta$ is a vector of coefficients for each predictor variable, and $b$ is the scalar bias). The algorithm minimizes an objective function using a dual stochastic gradient descent technique (see details below). We adopted the SVM-r linear regression approach as it is known to be well suited for high-dimensional and sparse data such as ours[34–36]. The algorithm is formulated as a linear epsilon-insensitivity SVM regression based on a convex optimization problem set to minimize residuals between the response data and the predicted function values, to be less than epsilon ($\varepsilon$). To ensure a computationally simple solution for the optimization problem that utilizes a predictor function that depends solely on the support vectors with $\beta$ parameters that can be described completely as a linear combination of the training observations[35,37,72], the linear function is expressed in its Lagrange dual formulation (Eq. 2):

$$F(x) = \sum_{n=1}^{N} (a_n - a_n^*)(x_n'x) + b \tag{2}$$

where $x_n'$ is the predictor data consisting of N multivariate observations. The $\beta$ parameter-values for the training observations are determined by (Eq. 3):

$$\beta = \sum_{n=1}^{N} (a_n - a_n^*) x_n'. \tag{3}$$

In this form, we can minimize the loss function (Eq. 4):

$$L(a) = \frac{1}{2} \sum_{i=1}^{N} \sum_{j=1}^{N} (a_i - a_i^*)(a_j - a_j^*) x_i' x_j + \varepsilon \sum_{i=1}^{N} (a_i + a_i^*) + \sum_{i=1}^{N} y_i(a_i^* - a_i) \tag{4}$$

constrained by: $\sum_{n=1}^{N}(a_n - a_n^*) = 0$, $\forall n : 0 \le a_n \le C$, $\forall n : 0 \le a_n^* \le C$ Where $a_n$ and $a_n^*$ are nonnegative multipliers for each observation $x_n$. $C$ is a numeric term that controls the penalty imposed on observations the lie outside the bounds of $\varepsilon$. It regularizes the model training to help prevent overfitting. To achieve optimal solutions, the linear SVM-r are constrained by the Karush-Kuhn-Tucker complementary conditions[33,35]:

$$\forall n : a_n(\varepsilon + \xi_n - y_n + x_n'\beta + b) = 0,$$

$$\forall n : a_n^*(\varepsilon + \xi_n^* + y_n - x_n'\beta - b) = 0,$$

$$\forall n : \xi_n(C - a_n) = 0, \forall n : \xi_n^*(C - a_n^*) = 0$$

Note that the formulation as outlined is based on the objective function (Eq. 5):

$$J(\beta) = \frac{1}{2}\beta'\beta + C\sum_{n=1}^{N}(\xi_n + \xi_n^*) \tag{5}$$

where $\xi_n$ and $\xi_n^*$ are slack variables that ensure feasible constraints for each observation[73]. To cross-validate the SVM-r model, we randomly partitioned the time-aligned pD predictor data matrix and the bladder pressure response vector acquired during filling and emptying of the bladder into training and testing subsets. The linear SVM-r model was derived by utilizing 80 % (1004 time-instant observations) of the randomly allocated pD data frames and bladder pressure data for training (Fig. 4B). We validated the model utilizing the remaining 20 % (250 time-instant observations) untrained test data subset to reconstruct the bladder pressure (BP) response. To generate the predicted and actual bladder pressure curves, the randomized time instants associated with untrained fUSI data frames and acquired bladder pressure were stored. The prediction performance of the SVM-r model was subsequently evaluated by calculating the mean-squared-error (MSE) – that represents deviations between the predicted bladder pressure values utilizing the model and the actual bladder pressure response values (Fig. 4D). The SVM-r analysis was performed separately for each patient. Finally, we projected the linear SVM-r $\beta$ vector of coefficients back into the 2D spinal cord image space to identify spinal cord regions with ΔSCBV signal that best capture the temporal changes of the bladder pressure. The coefficients represent weights that are assigned to each predictor variable and reflect the importance of each variable (i.e., pD spinal cord image pixel) for predicting the bladder pressure.

### Software analysis

All data pre- and post-processing and statistical analysis were performed using MATLAB Version 9.13.0.2193358 (R2023b).

### Reporting summary

Further information on research design is available in the Nature Portfolio Reporting Summary linked to this article.

## Data availability

The dataset for patient P2 is available at https://doi.org/10.6084/m9.figshare.29194454 Data for the remaining patients supporting the findings of this study are available from the corresponding authors upon request. These datasets are provided for non-commercial research use only. Requests that conflict with patient confidentiality, ethical guidelines, or consent agreements cannot be fulfilled. Source data are provided in this paper.

## Code availability

The code used for generating the activation maps are available https://doi.org/10.6084/m9.figshare.29198729

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

## Acknowledgements

We thank the participants that made this study possible. This work was supported by "The USC Neurorestoration Center" at the University of Southern California (D.J.L. and C.L.), "The Hellman Foundation" (V.N.C.) and the "Marlan and Rosemary Bourns College of Engineering" at the University of California, Riverside through start-up funding (V.N.C.).

## Author contributions

K.A.A., D.J.L., E.I.K., V.R.E., C.L., and V.N.C. conceived the study; D.J.L. performed the spinal cord surgeries; E.I.K. performed the urodynamic experiments; D.J.L., A.A., Y.T.L., K.W., W.C. and S.I. acquired the functional ultrasound data; K.A.A. performed the functional ultrasound data processing and the statistical and machine learning decoding analysis; K.A.A., D.J.L., and V.N.C. drafted the manuscript with substantial contribution from S.S., E.I.K., J.R., Y.T.L., V.R.E., and C.L.; All authors edited and approved the final version of the manuscript; C.L. and V.N.C. supervised the research.

## Competing interests

The authors declare no competing interests.
