## [Transparent Peer Review file · Nature Communications]

Human spinal cord activation during filling and emptying of the bladder

Corresponding Author: Dr Vasileios Christopoulos

Version 0:

Reviewer comments:

Reviewer #1

(Remarks to the Author)

Agyeman et al., following their previous proof-of-concept demonstration of human spinal cord imaging, now provide the first demonstration of human spinal hemodynamic signals strongly correlated with bladder pressure. This subject is highly interesting as it opens new perspectives for brain-machine interfaces. In this context, functional ultrasound (fUS) imaging is highly promising and specifically adapted to the sensitive readout of slow physical signals, such as bladder pressure sensation, where the temporal delay introduced by the neurovascular coupling is negligible. The manuscript is well-written and the illustrations are adequate. However, I have several concerns.

Major Concern:

In a highly intriguing way, the fUS signal from tissue dorsal and lateral of the perceived limits of the spinal cord also shows significant correlation with changes in bladder pressure. In patients P3 and P4, in all figures, this extra-spinal signal contributes maybe even more to the ability to detect changes in bladder pressure than the signal from the spinal cord proper. This raises the concern that instead of detecting signals of neuronal origin, the source of this extra-spinal fUS signal could be an artifact. One possibility is a movement artifact caused by the change in body shape of the prone-lying patient during mechanical bladder filling and emptying. The deformations from the bladder region might be directly transmitted through the sacral and lumbar spinal cord to the detection region. How can the authors exclude this possibility? Alternatively, this signal could come from operation wound bleeding, stronger in the first, filling phase of the experiment. If not an artifact, what is the source of this extra-spinal signal?

Minor Concerns:

The Abstract must be better balanced, actually it's 6 phrases Introduction, 1 phrase of methods and only one phrase of results.

Please show also the recorded 5 mins baseline data.

The authors state that Figures 5 and 6 show the fUS signal detecting bladder pressure "even when information about the bladder pressure is not available." This is only partly true: while the actual pressure value is not known, the machine learning algorithm is trained with the knowledge of being in a filling or emptying phase, so knowledge is used about the direction of pressure change.

The authors state that "the hemodynamic signal within the T10 area encodes bladder pressure", while "correlated with" might be more precise.

The reference list must be updated, as several preprints are now published.

Reviewer #2

(Remarks to the Author)

The coordination between bladder pressure and neuronal activity ensures micturition occurs under socially and environmentally appropriate conditions. This neuronal activity leads to localized changes in blood flow (neurovascular coupling) as the bladder fills and empties. Here, the authors demonstrate in human patients that hemodynamic signal within the T10 area somehow "encodes" bladder pressure. To do this, the authors show that Doppler ultrasound (fUSI) can detect spinal cord regions where the hemodynamic signal is highly correlated with bladder pressure. This signal, presumably, is used by higher brain centers to control voiding. This is a first in human demonstration and thus significant. A limitation is that the main control mechanism is located at a different location in the cord, as the authors acknowledge. This is done since this is opportunistic imaging for a procedure unrelated to the study. Nevertheless, the understanding could help design neuromodulation therapies to restore function in various pathologies. With the appropriate responses to the points raised, it should be published as this is a first in the human measurement of spinal cord fUSI.

Specific Comments

An image or schematic of the vascularization of the lumbar spinal cord in the imaging plane would help interpret the images presented. It could be in Figure 1A.

Figure 1C: the bladder pressure recordings - how variable is this measurement between the different bladder-filling cycles for an individual patient? What is plotted here?

Figure 3: In Fig 3A, when there is no color representation for the location of a vessel from the PD, for the correlation coefficient, what does this mean? Is there a threshold for the coloration for the correlation coefficient - between the bladder pressure and Δ SCBV (or is this what line 495 describes?). If the latter, how was this threshold chosen? (and please include in figure caption).

Figure 3: The colours in the caption of Figure 3C are challenging to distinguish. Also, the signal changes for each patient are shown for the PD but not for the bladder pressure. Could a figure similar to 3B also be displayed to demonstrate the change in the bladder pressure? Is that the data in Figure 1C? If so, the largest pressure differential invokes the smallest Peak pD signal change. Even though one would suspect this is highly variable between patients, at least this should be commented on. In the discussion, mention the impact of imaging different planes for each patient and how this may contribute to the variability seen in signal change.

I 201: Why were the hold times excluded? What was the purpose of the hold periods in the experimental design, and why were the times chosen? More broadly, how were the times in Figure 1B selected?

I 443: The Iconeus One is typically associated with brain imaging. What were the specific modifications made to do this imaging? What is the bandwidth of the transducer used? Was the probe provided by Iconeus (model number)? What were the acoustic amplitudes and intensities of the fUSI sequence? Technical details that would aid in reproducibility should be included in the methods section.

I 472: Are these implemented by the vendor, or are these implemented by the group? If the latter, have the authors published on this? How do the authors know that high frequencies correspond to noise?

I 487: How much did the baseline vary?

I 505: Can we use the term "time intervals" instead of epochs, which have specific meaning in machine learning?

I 542: I don't understand the basis of this assumption. Please justify.

I 554: Validation: Although this 80/20 training/testing is typical, please specifically state the number of datasets used in tandem. It is a bit unclear.

Reference 47 can be updated to the published paper (Science Translational Medicine doi: 10.1126/scitranslmed.adj3143).

General comment: A comparison of Figs 3a,5a and 6a is warranted in the discussion. Why do these seem to have regions that have large differences in the spatial location of the "weighted pixels" and the activation maps of the correlations (3a)? Wouldn't one expect these to be similar?

Some discussion is also needed on how robust the findings are, both in terms of the measurement technique itself (how does anesthesia impact this, different imaging plane for imaging and spine interference) and the populations of patients examined (failed back surgery syndrome)- whether the observed data correlations apply to this population only or are broader. Finally, the findings should be discussed more broadly in terms of what is known about the activity of the spinal cord during urodynamically-controlled bladder filling and emptying (e.g. <https://doi.org/10.1016/j.expneurol.2019.113033>). Finally, most data are not quantitative (% change or normalized), but the non-normalized data reported in an appendix would add value.

Reviewer #3

(Remarks to the Author)

The research presented in the article "Human spinal cord activation during filling and emptying of the bladder" by Agyeman et al. is an original study and well-written manuscript with potential impacts on patients. The study aims to map the hemodynamic response of the spinal cord - as measured by functional ultrasound imaging - correlated with bladder pressure in humans, potentially advancing understanding and treatment of urinary dysfunctions. However, several concerns need to be addressed to enhance the robustness and clarity of the findings.

First, while the authors claim that their work is unique compared to existing studies on spinal cord stimulation in humans, one could argue that artificially filling and emptying the bladder is a form of external stimulation. Despite this, the study's novelty in applying this approach to humans is commendable.

One strong concern is that the activation images presented here could be more convincing. The activation and deactivation blobs are poorly defined and spread out, with many activations outside the delineated spinal cord area. If one of the paper's goals is to demonstrate the use of functional ultrasound imaging (fUSI) to map activation during bladder filling and emptying, the anatomical localization of the activated areas should be more thoroughly discussed and cross-validated.

The setup implies that the probe is not fixed to the patient, suggesting that any patient movement or movement resulting from bladder emptying will shift the patient's position relative to the probe. The authors mention motion correction but do not provide details or examples of such corrections. This is concerning, as motion artifacts can lead to artificially correlated spots in the images, as observed here.

The correlation coefficients in the maps are sometimes saturated at +1 / -1. This is highly unusual, how is this possible in the given context to have such perfectly correlated pixels?

The authors mention using False Discovery Rate (FDR) correction but do not provide the method used or the statistical justification for it. Similarly, they use a correlation threshold of 0.35 without any methodology or justification. Applying a low-pass filter will also modify the degrees of freedom of the problem and impact the statistics and correlation values and thresholding strategy.

There are signs of circular reasoning in the results section: using correlation maps to find activated areas, plotting signals in those areas, and then measuring—and discussing—correlation values. This approach will lead to highly correlated values by construction and, as such, any statistical observations on this correlation measurement will be biased towards high values.

It is unclear what question the "machine learning" part is addressing. Please avoid using this vague label in the first part of the manuscript. It seems to be about localizing pixels related to "emptying versus filling." How is this more useful than the correlation maps for this? Why not rather try to detect - in real time or using few previous frames - if the bladder is full or empty using all pixels simultaneously, as proposed in the clinical context discussion? Similarly, in that context, why focus on the dynamic "emptying/filling" rather than the steady state "empty/full"?

The SVM classifier is configured to use the acquisition frames as independent training sets with validation based on remaining frames, but successive frames are highly correlated since they come from the same dataset and acquisition. This approach does not seem to provide any additional insights and will likely result in the same observations as those made from the correlation maps. Moreover, using a PCA for data reduction will likely further mix pixels and temporal samples during the projection step, making the new data non-independent in time. This further complicates the validity of using frames as independent samples for training and validating the SVM.

In my view, the study has significant strengths which should be better used to focus on:

- Validation of the technique and observed signal with strong anatomical and functional cross-validation.
- Application of a classifier within the proposed clinical context of urinary dysfunctions by detecting whereas the bladder is full.

On this last point, bladder size could also be measured non-invasively using ultrasound and could be used to generate a notification on the user's phone or watch. How is the proposed approach better suited for this task, as it requires reading this signal indirectly using an invasive technique which corresponds only to half of the problem since it does not also stimulate the brain?

By addressing these issues, the authors could significantly improve the robustness and clarity of their findings and manuscript.

Specific Comments:

- The abstract and introduction should mention that fUSI is invasive.
- In Figure 3, do not rescale deltaSCBV signals.
- Line 396: Correct the typo "perfectly about the dura."
- Line 453: Clarify "Trace 3" and "Track 3" with further details.
- Line 562: Define what constitutes a reasonable and unreasonable request.

Version 1:

Reviewer comments:

Reviewer #2

(Remarks to the Author)

The change in the manuscript's focus made it more compelling. Adding the baseline removes several concerns, and the new machine learning approach to predict bladder pressure makes more sense, providing better clarity and more robust conclusions. I recommend the publication of the manuscript.

Comments

I do not see the non-normalized pressure patient data in Figure S2. The non-normalized data should be reported in the appendix. For example, what are the intravesical bladder pressure recordings? (cmH₂O or whatever unit is used) - to gauge the range of physical values based on the % percentage change from baseline. Another reason I ask is that the $t=0$ min seems greater than the baseline? (Fig 5A compared to Fig 1A for all patients). This is considered a minor point.

I recommend including some of the text in the reply to the major concern of reviewer one in the manuscript. Indeed, the correlation coefficients for P3 and P4 look strong outside the dorsal surface, raising the question. In other words, in lines 234-240, talk about Fig S3 in more detail - that leads to the claim that extra-spinal signal is related to the blood supply of the spinal cord itself.

Reviewer #3

(Remarks to the Author)

Thank you for the revised manuscript and for addressing the points raised in the previous review. The revisions improve several aspects of the work, but certain critical concerns remain only partially resolved and require further attention.

Major Concerns:

Motion artifacts remain a significant concern. The correlation between signals inside and outside the spinal cord raises the possibility of global motion effects rather than localized neural activity. To strengthen their claim, I recommend the inclusion of Doppler movies or quantitative displacement estimates from the motion correction step. These additions would provide stronger and definitive evidence to substantiate the argument that the observed signals are free from motion-induced artifacts.

The lack of probe fixation still raises methodological questions, particularly in comparison to cited related studies. Non-human primate experiments frequently use probe inserted in some chambers, neonatal studies rely on probes attached to a headcap, and intraoperative brain imaging use a mechanical arm to stabilize imaging probes. In those cases, either the probe is fixed to the head or both the probe and the head are fixed. While anesthesia may reduce patient movement, the absence of a fixation mechanism introduces the potential for subtle but systematic artifacts such as bladder filling.

The issue of independence between training and testing datasets in the SVM analysis remains inadequately addressed despite the proposed change of strategy. The low-frequency nature of bladder filling signals inherently creates strong inter frames temporal correlations, making successive frames dependent and challenging the validity of the model's generalization. The authors should address this limitation explicitly in the manuscript.

The manuscript would benefit from a more comprehensive discussion of all those limitations. The authors defend their methodological choices, but the manuscript itself should transparently acknowledge these challenges. For instance, discussing the potential implications of motion correction, probe stabilization, and strong temporal correlations in the SVM analysis would provide a more balanced and critical evaluation of the study.

Minor Suggestions:

The saturation visible in the correlation maps, even if not indicative of exact $+1/-1$ values, can be visually misleading. Are the data capped to a specific threshold or not?

The explanation provided for the use of FDR correction is appreciated but could be more straightforward. Citing previous literature is not a particularly strong methodological argument.

While the authors have made substantial progress in addressing previous concerns, critical issues related to motion artifacts and the SVM analysis require further attention.

The inclusion of a broader discussion of limitations would also enhance the manuscript's transparency and credibility.

I appreciate the authors thoughtful responses thus far and encourage further revisions to resolve these remaining concerns. Pending these revisions, I would support the manuscript acceptance.

Version 2:

Reviewer comments:

Reviewer #2

(Remarks to the Author)

The authors have provided compelling evidence and videos to demonstrate that the observed activation reflects localized spinal cord activity, rather than motion or other artifacts. The correlation threshold was explained, and the methods were better clarified. The limitations and clinical challenges have been significantly expanded.

Video SV1: Maybe have arrows to the most apparent motion artifacts in both videos, and somehow quantify the reduction in the motion fluctuations.

Reviewer #3

(Remarks to the Author)

The authors have thoroughly addressed all the concerns raised. I commend their efforts in implementing the requested changes and their important and valuable contribution to the field.

REVIEWER COMMENTS

Reviewer #1 (Remarks to the Author):

Agyeman et al., following their previous proof-of-concept demonstration of human spinal cord imaging, now provide the first demonstration of human spinal hemodynamic signals strongly correlated with bladder pressure. This subject is highly interesting as it opens new perspectives for brain-machine interfaces. In this context, functional ultrasound (fUS) imaging is highly promising and specifically adapted to the sensitive readout of slow physical signals, such as bladder pressure sensation, where the temporal delay introduced by the neurovascular coupling is negligible. The manuscript is well-written and the illustrations are adequate. However, I have several concerns.

We are pleased that the reviewer finds our manuscript interesting and are grateful for their effort to provide remarks and suggestions that we believe significantly improve our work. Accordingly, we have made every effort to address all the major and minor comments as outlined below.

Major Concern:

In a highly intriguing way, the fUS signal from tissue dorsal and lateral of the perceived limits of the spinal cord also shows significant correlation with changes in bladder pressure. In patients P3 and P4, in all figures, this extra-spinal signal contributes maybe even more to the ability to detect changes in bladder pressure than the signal from the spinal cord proper. This raises the concern that instead of detecting signals of neuronal origin, the source of this extra-spinal fUS signal could be an artifact. One possibility is a movement artifact caused by the change in body shape of the prone-lying patient during mechanical bladder filling and emptying. The deformations from the bladder region might be directly transmitted through the sacral and lumbar spinal cord to the detection region. How can the authors exclude this possibility? Alternatively, this signal could come from operation wound bleeding, stronger in the first, filling phase of the experiment. If not an artifact, what is the source of this extra-spinal signal?

Thank you for the important observation and questions. While the reviewer raises valid concerns, further analysis and review of the surgical procedure and fUSI recordings in the operating room suggest that the extra-spinal fUSI signal is not an artifact.

The observed activations derived from the correlation analysis that extend beyond the highlighted (white discontinuous lines) dorsal surface bounds, we believe indicate that neural signals associated with bladder function may modulate hemodynamic activity in regions adjacent to the gray matter of the spinal cord. It is also likely that the activation detected in vessels outside the dorsal column may be attributed to their role in

supplying blood to the vasculature within the gray matter. Note that we have also observed spinal cord activation outside the gray matter in our recent epidural spinal cord stimulation study (Agyeman et al., Neuron, 2024). Furthermore, if bladder manipulation (i.e., filling and emptying) induced severe motion in the spinal cord, we would expect the entire power Doppler (pD) image to be correlated with bladder pressure, as any motion would likely cause the pD image to shift uniformly like a rigid body. In such a case, all pixels across the image would show a strong correlation with bladder pressure. However, in our study, we observed that only specific regions of the spinal cord were correlated with bladder pressure, indicating localized functional activation rather than global motion. Importantly, we applied motion correction techniques to eliminate motion artifacts, further ensuring that the observed correlations are due to neural activity rather than mechanical movement of the spinal cord induced by the bladder, lungs and other neighboring organs (see Figure S1).

However, to address the reviewer's concerns, we performed further analysis whereby we isolated the extra-spinal signal and compared the temporal dynamics of the mean change in SCBV signal over the activated regions outside the dorsal surface to the activated signal within the spinal bounds. The results showed a consistent increase and decrease in %pD signal change during both bladder filling and emptying, occurring within as well as outside the dorsal bounds of the activated regions of the spinal cord (Figure S3). This leads us to believe that the extra-spinal signal is related to the blood supply of the spinal cord itself. The intra-spinal perfusion is fed by the extra-spinal arteries. Increased activation would be expected if there is increased intra-spinal blood flow and may be different on the left versus right depending on activation. Thus, it is unlikely to be an artifact. Additionally, the activation is within the dura and is very unlikely to be bleeding or extravasation. Indeed, the deformations from the bladder region might be directly transmitted through the sacral and lumbar spinal cord and the increase in flow of the extra-spinal arteries/ vessels may be associated with these changes. More specifically, extra-spinal vessel activation could be associated with more caudal activity. Finally, we added a supplementary figure in the revised manuscript showing that the motion correction algorithms used in our study have successfully removed motion artifacts (Figure S1). We have updated the Methods and Results section of the manuscript to highlight these points (Lines 234 – 240).

Minor Concerns:

The Abstract must be better balanced, actually it's is 6 phrases Introduction, 1 phrase of methods and only one phrase of results.

Thank you for the suggestion. We have revised the abstract based on the suggestion of the reviewer.

Please show also the recorded 5 mins baseline data.

Thank you for the suggestion. To address the reviewer's comment, we re-analyzed the spinal cord fUSI signals utilizing the entire 5 min of baseline signal as the reference activity in the correlation and machine learning analysis. We have subsequently revised the results of the %pD signal changes and the normalized SCBV changes to include responses over the entire baseline period (Figs. 3B, S1, S2, S3). (Lines 194 – 195)

The authors state that Figures 5 and 6 show the fUS signal detecting bladder pressure "even when information about the bladder pressure is not available." This is only partly true: while the actual pressure value is not known, the machine learning algorithm is trained with the knowledge of being in a filling or emptying phase, so knowledge is used about the direction of pressure change.

Thank you for the comments. While the reviewer makes an important observation, to address comments from other reviewers, we employed a different machine learning approach (regularized linear support vector machine regression analysis) to reconstruct the bladder pressure and to identify relevant spinal cord regions in the updated version of the manuscript. See Methods and Results section for details (Lines 245-316, 621-688, Figure 4, 5). Consequently, we have removed that sentence, and old Figures 5 and 6 are no longer included in the revised manuscript.

The authors state that "the hemodynamic signal within the T10 area encodes bladder pressure", while "correlated with" might be more precise.

Thank you for your observation. The reviewer makes an important point. In relation to the correlation analysis between the spinal cord %pD signal and bladder pressure signal, the reviewer is right. In the revised manuscript, we used a new machine learning strategy (support vector machine regression, SVM-r), the weighted pixels derived from the SVM-r beta parameter coefficients encode (i.e., predict/reconstruct) the bladder pressure (Lines 245-316, 621-688, Figure 4, 5)

The reference list must be updated, as several preprints are now published.

We updated the reference list included the new publications

Reviewer #2 (Remarks to the Author):

The coordination between bladder pressure and neuronal activity ensures micturition occurs under socially and environmentally appropriate conditions. This neuronal activity leads to localized changes in blood flow (neurovascular coupling) as the bladder fills and empties. Here, the authors demonstrate in human patients that hemodynamic signal within the T10 area somehow "encodes" bladder pressure. To do this, the authors show that Doppler ultrasound (fUSI) can detect spinal cord regions where the hemodynamic signal is highly correlated with bladder pressure. This signal, presumably, is used by higher brain centers to control voiding. This is a first in human demonstration and thus significant. A limitation is that the main control mechanism is located at a different location in the cord, as the authors acknowledge. This is done since this is opportunistic imaging for a procedure unrelated to the study. Nevertheless, the understanding could help design neuromodulation therapies to restore function in various pathologies. With the appropriate responses to the points raised, it should be published as this is a first in the human measurement of spinal cord fUSI.

Specific Comments

An image or schematic of the vascularization of the lumbar spinal cord in the imaging plane would help interpret the images presented. It could be in Figure 1A.

Thank you for the excellent suggestion. To address the reviewer's comment, we have updated the manuscript to include a spinal cord rendering showing pertinent anatomical structures and vascularization of the lumbar region of the fUSI plane (Figure 2A). Additionally, we have included indicators on the spinal cord pD vascular map images, indicating the dorsal funiculus, posterior median sulcus and posterior spinal arteries to further assist with interpreting the images (Figures 2B-E, 3A, 5D). We have amended the results section of the manuscript to reflect these edits (Lines 178 – 188).

Figure 1C: the bladder pressure recordings - how variable is this measurement between the different bladder-filling cycles for an individual patient? What is plotted here?

We need to point out here that we performed one cycle of bladder filling and emptying due to protocol and restrictions in the operating room that result in limited acquisition time. The total time of a single urodynamic cycle is approximately 26 minutes. On the other hand, while the general pattern of increase and decrease of the bladder pressure recordings during filling and emptying are consistent and similar across all patients, we observed some variability in the peak signal responses of the % bladder pressure changes across patients. To assess the variability across patients, we determined the percentage changes of the bladder pressure relative to bladder pressure signal

acquired during the first 30 s after starting to fill the bladder, for all 4 patients. The individual patients and the mean % bladder pressure changes across all patients (with the standard error indicated by the surrounding grey region) show gradual increase and decrease during the filling and emptying phases of the bladder, respectively (Figure 1D, left panel). We observed a mean peak bladder pressure signal change of 203.23 ± 29.65 % (Mean \pm SE) across all patients (Figure 1D, right panel). The individual peak bladder pressure signal changes for each patient are indicated by the color dots, blue, orange, yellow and purple for patients 1, 2, 3, and 4, respectively. We have revised the manuscript to include this new analysis (Lines 159 – 174).

Figure 3: In Fig 3A, when there is no color representation for to location of a vessel from the pD, for the correlation coefficient, what does this mean? Is there a threshold for the coloration for the correlation coefficient - between the bladder pressure and Δ SCBV (or is this what line 495 describes?). If the latter, how was this threshold chosen? (and please include in figure caption).

Thank you for the question. While we performed pixelwise analysis to determine a coefficient for each pixel, we set correlation coefficient thresholds to capture the significant and top 5% positive and negative correlates of spinal cord pixels to the bladder pressure [$p < 0.01$, after FDR correction, ($r > 0.43$, z-score > 0.49 , positive correlates) and ($r < -0.46$, z-score < -0.53 , negative correlates)]. The top-ranked pixels are depicted as the activation overlay color-regions over the mean grey-scale background baseline pD signal. We selected this threshold to avoid overcrowding the activation map and to better visualize the areas where the pD signal is strongly correlated with the bladder pressure. The corresponding mean %pD signal changes in SCVB are derived by averaging the signal over the significant and top-ranked (5%) spinal cord pixels (Figure 3). Note that previous studies followed the same approach by setting a threshold to generate the activation maps in functional neuroimaging. For instance, a recent fUSI study used a threshold of ($r > 0.41$) to generate the activation map that illustrates the correlation between the pD signal and the stimulus (sensory and motor tasks) temporal patterns (Imbault *et al.*, 2017, *Scientific report*). Similarly, a recent study from our team used a threshold of 10% to visualize the “most heavily weighted pixels” generated by the classwise principal component analysis (cPCA). These pixels contain strong information that can predict the direction of the intended movement (i.e., left vs. right) before the animal performs the action (Norman *et al.*, 2021, *Neuron*)

To address the question and further clarify the reviewer’s comments, we have revised the manuscript in the Methods and Results sections to emphasize this point and provide further details about the statistical methods employed and how the threshold was chosen (Lines 198 – 208, 614 – 617). Additionally, we have updated the figure caption per the reviewer’s suggestion (Figure 3).

Figure 3: The colours in the caption of Figure 3C are challenging to distinguish. Also, the signal changes for each patient are shown for the PD but not for the bladder pressure. Could a figure similar to 3B also be displayed to demonstrate the change in the bladder pressure? Is that the data in Figure 1C? If so, the largest pressure differential invokes the smallest Peak pD signal change. Even though one would suspect this is highly variable between patients, at least this should be commented on. In the discussion, mention the impact of imaging different planes for each patient and how this may contribute to the variability seen in signal change.

Thank you for the comments. To address the point about the caption for Figure 3C, we have revised the figure caption to highlight the different color bars (red, blue, and dark grey). The reviewer also made an important observation about presenting the bladder pressure changes as well. Accordingly, we determined the percentage changes of the bladder pressure recordings relative to the bladder pressure signal acquired during the first 30 s, for all 4 patients. The mean bladder pressure changes (with the standard error indicated by the surrounding grey region) and the individual patient bladder pressure changes show gradual increase and decrease during the filling and emptying phases of the bladder respectively (Figure 1D, left panel). We observed a mean peak bladder pressure signal change of 203.23 ± 29.65 % (Mean \pm SE) across all patients (Figure 1D, right panel). The individual peak bladder pressure signal changes for each patient are indicated by the color dots (blue, orange, yellow and purple for patients 1, 2, 3 and 4 respectively).

Regarding the question of whether we can assess a correlation between the peak of the pD signal and the peak of bladder pressure – i.e., largest pressure invokes smaller or larger pD signal – it is challenging to draw definitive conclusions given our sample size of four patients. We agree this is an important question, but it requires further investigation with a larger patient sample, which we can address in future studies. Also, the variability on the bladder pressure, it is something that we expected given the natural variability in bladder capacity and compliance across people – e.g., some people will accommodate a larger volume of fluid in the bladder than other people. The variability in bladder pressures among the participants reflects this natural variability. In some ways, it would be more surprising if the pressures were the same across all patients. Finally, the variability on the magnitude of the pD signal can reflect different factors, such as the variability on the imaging plane, the variability on the bladder capacity and compliance – i.e., some people need to void more often than others, and other. In the revision, we comment about the variability that we observe in the pD signal change across participants (see Lines 159 – 174, 473 – 477).

I 201: Why were the hold times excluded? What was the purpose of the hold periods in the experimental design, and why were the times chosen? More broadly, how were the times in Figure 1B selected?

Thank you for the comments. The reviewer makes an important observation. Although the hold times were excluded in the previous cPCA and SVM analysis, in the current version of the manuscript we do include the hold times to perform the linear SVM regression analysis (Fig. 4, 5). See the methods and results section for details. The hold period was implemented for two reasons:

1. Natural events during the human micturition cycle where humans maintain continence with a full bladder despite the urge to urinate.
2. A period to allow accommodation of rising bladder pressure as the bladder fills at a high rate.

The period of 90 seconds (i.e., 1.5 minutes) was chosen based on our previous experience allowing bladder pressure to equilibrate (*Kreydin et al., 2020, NeuroUrol Urodyn.*). The total duration of the experiment was chosen to simulate the natural progression of bladder filling during a typical micturition cycle. Rapid bladder filling was avoided, as it could trigger unwanted bladder contractions. On average, adults urinate every 3 to 4 hours. However, due to time constraints associated with the surgery, we were limited to conducting the urodynamic experiment approximately lasting no more than 30 minutes (5 minutes of baseline recordings + about 25 minutes of filling and emptying the bladder). We discussed the experimental protocol in the revision (see Lines 148 – 157).

I 443: The Iconeus One is typically associated with brain imaging. What were the specific modifications made to do this imaging? What is the bandwidth of the transducer used? Was the probe provided by Iconeus (model number)? What were the acoustic amplitudes and intensities of the fUSI sequence? Technical details that would aid in reproducibility should be included in the methods section.

Thank you for the questions. To address the reviewer's comments, we have included additional technical details relating to the Iconeus One system utilized in the study (Lines 560 – 573). We did not optimize the pulse sequence for the human spinal cord, since Iconeus One is a commercially available device and we do not have access to the underlying ultrasound sequence. In our recent publication; functional ultrasound imaging of the human spinal cord (*Agyeman et al. 2024, Neuron*), we acknowledged that it is important in the future to optimize the pulse sequence to the human spinal cord. In fact, we are working with Iconeus to generate a new pulse sequence that is optimized for the human spinal cord hemodynamics. Note that in the Neuron paper we showed that fUSI can predict the effects of epidural spinal cord stimulation on the spinal cord hemodynamics, although the pulse sequence is not optimized to the human spinal cord. Additionally, recent studies that image spinal cord of rodents used very similar sequences with the one that we used in our study (see for instance the work by *Claron et al., 2021, Pain*). Overall, the ability to reconstruct the bladder pressure with high accuracy and to detect spinal cord regions correlated with bladder filling and emptying reflect the robustness of fUSI technology, despite the fact that the pulse sequence is not

optimized to the human spinal cord. We have updated the manuscript to highlight these challenges and points in the Discussion section (Lines 496 – 502, 544 – 545, 556 – 560).

I 472: Are these implemented by the vendor, or are these implemented by the group? If the later, have the authors published on this? How do the authors know that high frequencies correspond to noise?

Thank you for the questions. The phase-correlation based sub-pixel motion registration and singular-value-decomposition based clutter filtering algorithms used to separate tissue motion signal from blood signal and to subsequently generate the relative pD signal intensity images are implemented by the Iconeus One System. We have revised this part to be clearer that these algorithms are implemented by the vendor, and we do not have access to them (Lines 580-583).

Regarding the high frequency fluctuations, these are often considered noise in functional neuroimaging based on several factors. The relevant physiological signals that are correlated with neural activity typically occur at low frequencies. The hemodynamic response that reflects changes in blood volume, generally appears between 0.01 to 0.1 Hz. This is similar to low frequency fluctuation that we observed in other neuroimaging studies like fMRI. Higher frequency fluctuations are often associated with noise and artifacts, such as motion, vibration from the scanner (i.e., electronics, power supply, probe, etc) and other non-physiological fluctuations. To address the reviewer's comments, we have included a supplementary figure in the revised manuscript showing the application of the motion correction algorithms to successfully remove the high frequency fluctuations (Figure. S1). We have updated the Methods section of the manuscript with additional detail to highlight these points (Lines 591-599).

I 487: How much did the baseline vary?

Great point. This is a very important question that we oversights in the original submission. In the revision, we computed the %pD signal changes for the first 5 min of baseline activity (i.e., before starting to fill the bladder). The analysis shows that the %pD signal in the activation regions of the spinal cord is relatively stable and does not vary much (Figure 3). Note that we have re-analyzed the spinal cord pD signal utilizing the entire 5 min of baseline activity as the reference in the correlation analysis (i.e., activation maps) and machine learning analysis (i.e., SVM-r). Additionally, to address the reviewer's question, we have subsequently revised the results of the %pD signal changes and the normalized SCBV changes to include the responses determined over the entire baseline period (Figures 3B, S1, S2, S3). (Lines 194 – 195, 212 – 213, 635 – 636).

I 505: Can we use the term "time intervals" instead of epochs, which have specific

meaning in machine learning?

Thank you for the suggestion. This phrase has been removed in the revision, since we implemented a new machine learning algorithm.

I 542: I don't understand the basis of this assumption. Please justify.

Thank you for the comments. We agree that the original description of how we determined the optimal amount of training data for the machine learning algorithm was unclear. However, in response to the comments from other reviewers, we have now adopted a different transductive machine learning approach. Specifically, in the revised manuscript, we utilized linear support vector machine regression to reconstruct bladder pressure and identify relevant spinal cord regions. See Methods and Results section for details (Figures 4, 5). Consequently, the assumption in line 542 is no longer applicable to the current machine learning analysis, and thus we have removed that sentence from the revised manuscript.

I 554: Validation: Although this 80/20 training/testing is typical, please specifically state the number of datasets used in tandem. It is a bit unclear.

Thank you for the suggestion. To cross-validate the SVM-r model training, we randomly partitioned the time-aligned fUSI predictor data matrix and the bladder pressure response vector into training and testing subsets. The linear SVM-r model was derived by utilizing 80 % (1004 time-point observations) of the randomly allocated pD predictor and bladder pressure response data subsets for training (Fig. 4B) for each patient's data separately. Note that the training and testing datasets are composed of randomly selected image-frames and pressure values acquired during the entire process of filling and emptying of the bladder. We then validated the model utilizing the remaining 20 % (250 time-point observations) untrained test data subset to predict the bladder pressure response and subsequently evaluated the decoding performance by calculating the mean-squared-error (MSE) between the actual and SVM-r predicted bladder pressures. Note that the corresponding time-points across filling and emptying of the bladder associated with the untrained testing fUSI and bladder pressure data sets were stored to allow reconstruction of the predicted and actual bladder pressure curves during validation.

It is important to point out that the current experimental design was limited to a single micturition cycle, preventing us from making predictions about future bladder pressure changes. Multiple micturition cycles would be required for *inductive* learning approaches capable of generalization. Consequently, we employed a transductive learning approach, focusing on reconstructing (i.e., predicting) bladder pressure from the current pD signal alone. In the future, we are planning to test the generalization capability of

fUSI technology. We have included more details about the validation data partitioning approach in the Methods and results sections, as well as in the figure captions (Fig. 4) (See lines 245 – 316, 621 – 688).

Reference 47 can be updated to the published paper (Science Translational Medicine doi: 10.1126/scitranslmed.adj3143).

We updated the reference list included the new publications

General comment: A comparison of Figs 3a,5a and 6a is warranted in the discussion. Why do these seem to have regions that have large differences in the spatial location of the "weighted pixels" and the activation maps of the correlations (3a)? Wouldn't one expect these to be similar?

Great point! In fact, this is not the first time that we see some differences between the correlation maps and the maps generated by a cPCA+LDA (or SVM) (i.e., the machine learning analysis). In our recent study in the human spinal cord stimulation (*Agyeman et al., 2024, Neuron,*), we performed a similar analysis – i.e., a) statistical parametric analysis looking into pixels where the pD signal is significantly higher during stimulation, and b) cPCA + LDA analysis to identify these pixels that accurately predict the effectiveness of a stimulation protocol to evoke hemodynamic changes in the spinal cord at the single-trial level. The results showed that the activated maps generated by the SPM analysis (i.e., pixels with high z-scores) and the cPCA + LDA machine learning algorithm (i.e., pixels with high decoder weights) showed some differences. The reason is that the SPM method performs a pixel-by-pixel analysis using only the average %pD signal change between the two classes (stimulation OFF vs. stimulation ON) to identify regions that exhibit the strongest hemodynamic signal induced by stimulation. On the other hand, the machine learning algorithm considers not only the amplitude of the %pD signal change but also the spatiotemporal patterns of the whole pD image to extract features in a lower dimensional space that best separate the two classes (see page 1714, left column, last paragraph in *Agyeman et al, 2024, Neuron*). Similarly, in the current study, the correlation analysis is performed separately for every pixel, whereas the SVM-r analysis consider the spatiotemporal interactions of all pixels in the pD image.

Akin to our recent *Neuron* publication, in the current study we anticipate differences between the two activation maps. The first analysis (pixel-wise correlation between the pD signal and bladder pressure) identifies regions where the pD signal correlates with bladder pressure, whereas the second analysis (SVM-r) identifies regions where the pD signal can predict bladder pressure. These differing objectives are expected to yield some differences between the activation maps. We revised the manuscript to point out

these reasons for the differences between the activation maps (see Lines 310 – 316).

Some discussion is also needed on how robust the findings are, both in terms of the measurement technique itself (how does anesthesia impact this, different imaging plane for imaging and spine interference) and the populations of patients examined (failed back surgery syndrome)- whether the observed data correlations apply to this population only or are broader. Finally, the findings should be discussed more broadly in terms of what is known about the activity of the spinal cord during urodynamically-controlled bladder filling and emptying (e.g. <https://doi.org/10.1016/j.expneurol.2019.113033>).

All these are excellent points that we have addressed in the revision.

1. **Different imaging plane:** In the original submission, we wrote. *“Although the imaging planes vary slightly across the 4 patients, this does not affect the spatiotemporal pattern of the hemodynamic signal in the bladder pressure-related regions. In fact, this highlights the strength and robustness of fUSI to overcome the potential to image different 2D slices of the spinal cord across patients”* (see Lines 473-478 in the revision).
2. **Anesthesia:** Here the reviewer brings up an important issue about anesthesia. It is known that anesthesia can affect local blood flow, leading to changes in blood perfusion. Since fUSI measures blood flow, alternations induced by anesthesia can affect the recordings. However, our study focuses on identifying how the hemodynamic signal changes with respect to the changes of the bladder state. Since the anesthesia is the same across the different stages of the bladder cycle, the confound is consistent, allowing us to attribute any observed differences in the hemodynamic signal to bladder filling and emptying. Additionally, baseline activity was recorded before the start of the bladder filling and emptying cycle, and the %pD signal was computed relative to this baseline, ensuring that changes were measured against a consistent reference point. Given that the anesthesia remains the same across surgery, it should not have a major impact on the %pD signal change. We addressed the limitation of the anesthesia on the revised manuscript (see Lines 481-495). Nevertheless, we believe that future experiments should be performed in awake participants who have laminectomies. Our team is working on this direction.
3. **Populations of patients examined:** The patients included in our study have chronic back pain, but do not present with any urinary system dysfunction. They are on medication solely for pain management and have no history or symptoms of urinary incontinence or other urological conditions. As a result, we do not expect their chronic pain to affect the study's findings related to micturition, as their urinary systems function normally (see Lines 491-495).

Finally, we discussed our findings with the results from previous studies on how the human spinal cord controls micturition (Lines 460 – 463).

Finally, most data are not quantitative (% change or normalized), but the non-normalized data reported in an appendix would add value.

Thank you for the suggestion. To address the reviewer's concern, we have included the unscaled mean %pD signal change curves across all patients in the revised manuscript (Figure S2).

Reviewer #3 (Remarks to the Author):

The research presented in the article "Human spinal cord activation during filling and emptying of the bladder" by Agyeman et al. is an original study and well-written manuscript with potential impacts on patients. The study aims to map the hemodynamic response of the spinal cord - as measured by functional ultrasound imaging - correlated with bladder pressure in humans, potentially advancing understanding and treatment of urinary dysfunctions.

However, several concerns need to be addressed to enhance the robustness and clarity of the findings.

First, while the authors claim that their work is unique compared to existing studies on spinal cord stimulation in humans, one could argue that artificially filling and emptying the bladder is a form of external stimulation. Despite this, the study's novelty in applying this approach to humans is commendable.

We are pleased that the reviewer finds our manuscript commendable and original. Their effort to provide remarks and suggestions that we strongly believe significantly improve our work is very much appreciated. Accordingly, we have made every effort to address all major and minor comments as outlined below. We agree that the artificial filling and emptying of the bladder can be viewed as a form of external stimulation of the spinal cord. However, this type of stimulation is different than the direct epidural spinal cord stimulation that our team recently published (*Agyeman et al. 2024, Neuron*) or the peripheral nerve stimulation studies, such as the one published by *Claron et al., 2021, Pain*, in rodents. Here, we assess how the spinal cord responds to a stimulation of an end-organ. Note that we revised the abstract so that we focus more on the novelty which is the characterization of the spinal cord hemodynamics during filling and emptying of the bladder.

One strong concern is that the activation images presented here could be more convincing. The activation and deactivation blobs are poorly defined and spread out, with many activations outside the delineated spinal cord area. If one of the paper's goals is to demonstrate the use of functional ultrasound imaging (fUSI) to map activation during bladder filling and emptying, the anatomical localization of the activated areas should be more thoroughly discussed and cross-validated.

Thank you for the observation and questions. The reviewer makes an excellent point about the localization of the activated spinal cord regions. To address the reviewer's comments, we have updated the manuscript to include a spinal cord rendering that shows pertinent anatomical structures and vascularization of the lumbar region of the fUSI plane (Figure 2A). Additionally, we have included indicators on the spinal cord pD

vascular map images, indicating the dorsal funiculus, posterior median sulcus and posterior spinal arteries to further assist with interpreting the images (Figures 2B-E, 3A, 5D) and have amended the Results section accordingly to reflect these edits (Lines 181 – 189).

Furthermore, while the reviewer raises valid concerns about observed activations outside the spinal cord dorsal surface, we believe that these activations possibly indicate that neural signals associated with bladder function may modulate hemodynamic activity in regions adjacent to the gray matter of the spinal cord. It is likely that the activation detected in vessels outside the dorsal column may be attributed to their role in supplying blood to the vasculature within the spinal cord gray matter. To address the reviewer's concerns, we performed further analysis whereby we isolated the extra-spinal signal and compared the temporal dynamics of the mean change in SCBV signal over the activated regions outside the dorsal surface to the activated signal within the spinal bounds. The results revealed consistent rise and fall of the %pD signal change during filling and emptying of the bladder for both (within and outside the dorsal bounds) activated spinal cord regions (Figure S3). This leads us to believe that the extra-spinal signal is related to the blood supply of the spinal cord itself. The intra-spinal perfusion is fed by the extra-spinal arteries. We have thus updated the Results and section of the manuscript to highlight these points (Lines 230-240).

In the revision, we performed a new analysis to identify spinal cord regions that predict bladder pressure changes. Specifically, we employed a regression-SVM model to identify spinal cord regions, where the power Doppler (pD) signal predicts the bladder pressure curve. Interestingly, these predictive regions are primarily located inside the dorsal surface of the spinal cord (Figure 5). These findings suggest that, while there are regions outside the dorsal surface where the pD signal correlates with bladder pressure, the regions within the dorsal surface of the spinal cord provide a more accurate reconstruction of the bladder pressure curve.

The setup implies that the probe is not fixed to the patient, suggesting that any patient movement or movement resulting from bladder emptying will shift the patient's position relative to the probe. The authors mention motion correction but do not provide details or examples of such corrections. This is concerning, as motion artifacts can lead to artificially correlated spots in the images, as observed here.

Thank you for the comments. The patients in our study are under deep anesthesia, which ensures they remain completely still throughout the procedure, eliminating concerns about body movement. Additionally, in the vast majority of previous studies involving non-human primates, clinical patients undergoing craniotomies, infants, even spinal cord imaging in rodents, the probe is not fixed to the subject's head/spinal cord, yet these studies (including our previous studies) have successfully removed motion artifacts following similar techniques that we used in our study. Furthermore, if bladder filling were to induce motion artifacts, we would expect a uniform shift in the power

Doppler (pD) image, resembling rigid body movement, where all pixels would correlate strongly with bladder pressure. In fact, we observed on the *raw* data strong motion artifacts that are (most likely) related to respiration – i.e., expansion and contraction of lungs during breathing. These artifacts have been successfully removed using techniques that are described in the manuscript (Figure S1) These techniques have been extensively incorporated in previous studies – including studies from our lab – to remove motion artifacts from fUSI data. A key indication that motion artifacts have been successfully corrected in the pD images is the ability of our machine learning approach to accurately identify specific regions of the spinal cord where the pD signal can reliably predict bladder pressure curves. If the pD images were predominantly affected by motion artifacts, such accurate bladder pressure predictions would be highly improbable. In the revision, we included a vascular map before and after motion correction from a representative patient (Figure S1) (see Lines 592-599).

The correlation coefficients in the maps are sometimes saturated at +1 / -1. This is highly unusual, how is this possible in the given context to have such perfectly correlated pixels?

The activation map shows only the significant top-ranked (5%) pixels where the pD signal is positively and negatively correlated to the bladder pressure. Therefore, it is expected that the correlation coefficients are very high. Since we only show the top 5%, some of the correlation coefficients in the maps are close to +1 or -1, but not exactly +1 or -1. This appearance is due to the colormap used in the activation maps, which can make these values seem saturated. Note that perfectly correlated pixels, with values of exactly +1 or -1, would appear as very dark red or dark blue, which do not actually show up on the maps. The observed strong correlations are below these extreme values, but the colormap can visually exaggerate their proximity to the limits. In the revision, we added a new figure (Figure 3D) that shows the mean of the top-ranked 5% correlation coefficients of the positive and negative correlations across subjects – as well as the individual mean top-ranked 5% correlation coefficients for each subject.

The authors mention using False Discovery Rate (FDR) correction but do not provide the method used or the statistical justification for it. Similarly, they use a correlation threshold of 0.35 without any methodology or justification. Applying a low-pass filter will also modify the degrees of freedom of the problem and impact the statistics and correlation values and thresholding strategy.

Great points. We used the Benjamini-Hochberg (BH) procedure for FDR correction (please see Line 624 – 628) in the revised manuscript. Also, we set the correlation threshold to capture the significant and top 5% positive and negative correlates, respectively to the bladder pressure. Note that we used $r >> 0.43$ and $r < -0.46$, (the top 5% on the revised manuscript), because these correspond to z-score > 0.49 and z-

score < -0.53 , respectively i.e., $p < 0.01$ with FDR correction, see lines 200 – 208, 610 – 612 in the revised manuscript). This methodology has been adopted by many studies in functional neuroimaging to generate activation maps. For instance, a recent fUSI study in human patients aimed to identify brain regions where the pD signal is correlated with motor task (e.g., move fingers, wrist, elbow, etc). The authors performed a similar correlation analysis to our work. The activation maps were built using a threshold of $r > 0.41$ (Imbault, 2017, *Scientific Reports*). Similarly, a recent study from our team used a threshold of 10% to visualize the “most heavily weighted pixels” generated by the classwise principal component analysis (cPCA). These pixels contain strong information that can predict the direction of the intended movement (i.e., left vs. right) before the animal performs the action (Norman, 2021, *Neuron*).

There are signs of circular reasoning in the results section: using correlation maps to find activated areas, plotting signals in those areas, and then measuring—and discussing—correlation values. This approach will lead to highly correlated values by construction and, as such, any statistical observations on this correlation measurement will be biased towards high values.

We understand the concern of the reviewer regarding the potential for circular reasoning. However, we adopted an approach that has been used in previous functional neuroimaging studies that aim to identify and visualize functionally active regions, as we discussed above. Specifically, we first perform a pixel-wise correlation analysis between the pD signal and bladder pressure to identify pixels that exhibit the strongest correlation, to generate the activation maps. The subsequent plotting of the average pD signal and bladder pressure from the highly correlated pixels is not intended to reinforce the initial correlation but rather to provide a clear visualization of the temporal pattern of pD signal changes alongside bladder pressure changes. This step allows for a better understanding of how the pD signal evolves over time in regions already identified as functionally related to bladder activity. While the correlation analysis identifies the most relevant pixels, the purpose of plotting the average signal is purely illustrative, enabling us to present the dynamics of the pD signal and bladder pressure in a way that is more accessible to the reader. We adopted the same approach in our previous studies in non-human primates (Norman et al., *Neuron*, 2021) and humans (Agyeman et al., *Neuron*, 2024).

It is unclear what question the "machine learning" part is addressing. Please avoid using this vague label in the first part of the manuscript. It seems to be about localizing pixels related to "emptying versus filling." How is this more useful than the correlation maps for this? Why not rather try to detect - in real time or using few previous frames - if the bladder is full or empty using all pixels simultaneously, as proposed in the clinical context discussion?

Similarly, in that context, why focus on the dynamic "emptying/filling" rather than the

steady state "empty/full"?

The SVM classifier is configured to use the acquisition frames as independent training sets with validation based on remaining frames, but successive frames are highly correlated since they come from the same dataset and acquisition. This approach does not seem to provide any additional insights and will likely result in the same observations as those made from the correlation maps. Moreover, using a PCA for data reduction will likely further mix pixels and temporal samples during the projection step, making the new data non-independent in time. This further complicates the validity of using frames as independent samples for training and validating the SVM.

In my view, the study has significant strengths which should be better used to focus on:

- Validation of the technique and observed signal with strong anatomical and functional cross-validation.
- Application of a classifier within the proposed clinical context of urinary dysfunctions by detecting whereas the bladder is full.

Excellent point and assessment! We sincerely thank the reviewer for this valuable suggestion, which we implemented in the revision. This feedback prompted us to think more critically about how to better apply the machine learning technique, ultimately strengthening the manuscript. So, in the revision, we used a transductive learning approach, focusing on reconstructing bladder pressure from the current pD signal alone. Note that the current experimental design was limited to a single micturition cycle, preventing us from making predictions about future bladder pressure changes. Hence, we introduced a new machine learning approach (regularized linear support vector machine regression analysis) to reconstruct the bladder pressure and to identify relevant spinal cord regions. See the Methods (see Lines 621-688) and Results (see Lines 245-316) section for details (Figures 4, 5). The new approach now allows us to predict the bladder pressure with high accuracy utilizing untrain whole spinal cord pD image frames for validation after training a linear SVM regression model on a randomized subset of pD signal acquired during filling and emptying of the bladder for each patient.

On this last point, bladder size could also be measured non-invasively using ultrasound and could be used to generate a notification on the user's phone or watch. How is the proposed approach better suited for this task, as it requires reading this signal indirectly using an invasive technique which corresponds only to half of the problem since it does not also stimulate the brain?

We acknowledge the potential of non-invasive ultrasound methods for estimating bladder size and triggering notifications through mobile devices. However, our approach goes beyond simple bladder size monitoring. It provides important insights into how

bladder pressure correlates with spinal cord activity. Specifically, our method characterizes the hemodynamic response in the spinal cord during urodynamically-controlled micturition and identifies spinal cord regions where the hemodynamic signal can predict bladder pressure. This is particularly valuable for understanding the spinal cord mechanisms involved in normal bladder control. It is also essential for identifying and characterizing abnormalities in patients with urinary incontinence. Our study sets the basis to start exploring the mechanisms of micturition in the human spinal cord. In the end, if you do not understand how a system works, you will not be able to fix it when it fails. We do not claim that our study addresses all the questions on how the spinal cord controls micturition. But it is the first study that introduces the fUSI technology as a tool for exploring the mechanisms of micturition in the human spinal cord.

Additionally, revealing neuroactivation patterns in the spinal cord during micturition could help optimize neuromodulatory therapies, such as transcutaneous spinal cord stimulation, for treating urinary incontinence. In relation to the potential use of fUSI in developing spinal cord-machine interfaces for bladder restoration, fUSI could detect impending voiding events, transmitting this information to the brain and using electrodes to stimulate sensory areas so patients regain the sensation of urgency to void. Patients with paralysis frequently express that their ideal solution would not only restore bladder control but also allow them to perceive the urgency to void. Simply monitoring bladder activity with a non-invasive system is not enough. Much like brain-computer interfaces for motor control, restoring motor function is just as important as restoring sensory function (i.e., the ability to feel when touching an object) for patients with paralysis who have lost control of their limbs. In the revision, we discussed why a non-invasive monitoring system is not enough to understand the mechanisms of micturition. We also discuss the advantage of our approach compared to non-invasive ultrasound approach for monitoring the bladder state in the revision (see Lines 427-439).

By addressing these issues, the authors could significantly improve the robustness and clarity of their findings and manuscript.

We would like to thank the reviewer for the comments and suggestions. Especially, their suggestion for using a different machine learning approach that can be used to reconstruct the bladder pressure, significantly improved our manuscript.

Specific Comments:

- The abstract and introduction should mention that fUSI is invasive.

We have added it in the revised manuscript (Lines 62 and 116).

- In Figure 3, do not rescale deltaSCBV signals.

Thank you for the suggestion. To address the reviewer's concern, we have included the unscaled mean change in SCBV (mean %pD signal change) curves across all patients

in the revised manuscript (Figures. S2 and S3A). While we acknowledge the reviewer's point, we also added the normalized %pD signal change curves and mean bladder pressure signal across patients to highlight the strong correlation between the signals observed (Figure 3B). We have revised the manuscript to reflect these changes (Lines 210-228).

- Line 396: Correct the typo "perfectly abut the dura."

Thank you for the reviewer's observation. We have edited "perfectly abut the dura" to "abut perfectly to the dura" – indicating that the fUSI probe did not touch the surface of the spinal cord (Line 470).

- Line 453: Clarify "Trace 3" and "Track 3" with further details.

We removed Track 3 from the manuscript. We referred to the 510 (k) FDA safety requirement (Line 558-559).

- Line 562: Define what constitutes a reasonable and unreasonable request.

Thank you for your question. A reasonable request means that the dataset and analysis scripts (Matlab scripts) from our study can be shared with other researchers strictly for scientific purposes. The data is not available for commercial use. Additionally, we cannot fulfill requests that would violate patient confidentiality, ethical guidelines, or the terms of patient consent agreements (please see lines 698-701).

REVIEWER COMMENTS

Reviewer #2 (Remarks to the Author):

The change in the manuscript's focus made it more compelling. Adding the baseline removes several concerns, and the new machine learning approach to predict bladder pressure makes more sense, providing better clarity and more robust conclusions. I recommend the publication of the manuscript.

We are pleased that the reviewer finds our updates have made the manuscript more compelling, robust, and worthy of publication. Below, we have carefully addressed all additional comments to further strengthen our work.

Comments

I do not see the non-normalized pressure patient data in Figure S2. The non-normalized data should be reported in the appendix. For example, what are the intravesical bladder pressure recordings? (cmH₂O or whatever unit is used) - to gauge the range of physical values based on the % percentage change from baseline. Another reason I ask is that the t=0min seems greater than the baseline? (Fig 5A compared to Fig 1A for all patients). This is considered a minor point.

Thank you for your observation. It is an unfortunate omission. In accordance with the reviewer's suggestion, we have included the non-normalized patient pressure data (in the recording units, cmH₂O) to the supplementary section (Fig. S2A). The individual patients' bladder pressure and the mean pressure (black curve) across all patients (with the standard error indicated by the surrounding grey region) are presented. We have thus updated the Results section of the manuscript to reflect the edits (Lines 168 – 170). Additionally, we would like to clarify that pressure recordings began at the start of bladder filling. As a result, we do not have baseline bladder pressure data (i.e., prior to initiating bladder filling).

I recommend including some of the text in the reply to the major concern of reviewer one in the manuscript. Indeed, the correlation coefficients for P3 and P4 look strong outside the dorsal surface, raising the question. In other words, in lines 234-240, talk about Fig S3 in more detail - that leads to the claim that extra-spinal signal is related to the blood supply of the spinal cord itself.

Thank you for the suggestion. Based on the reviewer's recommendation, we have expanded the results section to include some of the text from our response to add clarity to the manuscript (Lines 238 – 255).

Reviewer #3 (Remarks to the Author):

Thank you for the revised manuscript and for addressing the points raised in the previous review. The revisions improve several aspects of the work, but certain critical concerns remain only partially resolved and require further attention.

We sincerely appreciate your thoughtful feedback, and your recognition of the substantial progress made in our revisions. In response to your comments, we have carefully addressed the critical issues related to motion artifacts and the SVM analysis, ensuring a thorough discussion of these aspects in the manuscript. Additionally, we have expanded the limitations section to enhance the transparency and credibility of our study. We believe that these revisions have further strengthened the manuscript, and we thank you for your constructive input throughout the review process.

Major Concerns:

Motion artifacts remain a significant concern. The correlation between signals inside and outside the spinal cord raises the possibility of global motion effects rather than localized neural activity. To strengthen their claim, I recommend the inclusion of Doppler movies or quantitative displacement estimates from the motion correction step. These additions would provide stronger and definitive evidence to substantiate the argument that the observed signals are free from motion-induced artifacts.

Thank you for the comments. To strengthen our assertion that the pD signal and bladder pressure correlation activations found outside the dorsal regions of the spinal cord are unlikely to be related to artifact effects, we have included before and after motion correction pre-processing movies, as recommended by the reviewer to the Supplementary section of the revised manuscript (SV1). Motion artifacts can be observed in the unfiltered movie (sped-up, 20 frames per second) (SV1A). In contrast, after the motion artifact and high frequency pre-processing step, the pD signal is clearly stabilized and the motion fluctuations are eliminated in the filtered movie (sped-up, 20 frames per second) (SV1B). Note that the unfiltered and filtered movies represent the pD spinal cord activity acquired during the 5 min baseline period and the entire 6 min 40 s of the first bladder filling phase (11min 40s total), for a typical patient (patient 4).

To further ensure that the observed activation reflects localized spinal cord activity, we extracted mean Δ SCBV signals from a bladder pressure-correlated region and an adjacent non-correlated region across the total period of the experiment. (Fig. S3 C, D; patient P2). We found that the average percentage Δ SCBV curve from the bladder

pressure-correlated region (red curve) shows a gradual increase and decrease during bladder filling and emptying. In contrast, the Δ SCBV waveform from the adjacent non-correlated region (dark gray curve) remains flat throughout both the urodynamic experiment. These results indicate that the observed spinal cord activations are region-specific rather than global, with pD signal changes significantly correlated to bladder pressure in distinct spinal regions.

Note that for Figure S1, we used patient P1; for Figures S3C and S3D, we used patient P2; and for movie V1, we used patient P4. Since all these figures relate to motion artifact correction, we included different patients to strengthen our claim that the preprocessing algorithms used in this study effectively removed motion artifacts across all patients.

We have revised the Methods and Results section of the manuscript accordingly, to reflect these updates (SV1, Fig. S3 C, D) (Lines 238 – 255, 498 – 505, 636 – 643).

The lack of probe fixation still raises methodological questions, particularly in comparison to cited related studies. Non-human primate experiments frequently use probe inserted in some chambers, neonatal studies rely on probes attached to a headcap, and intraoperative brain imaging use a mechanical arm to stabilize imaging probes. In those cases, either the probe is fixed to the head or both the probe and the head are fixed. While anesthesia may reduce patient movement, the absence of a fixation mechanism introduces the potential for subtle but systematic artifacts such as bladder filling.

We appreciate the reviewer's comment regarding probe fixation. We realize that we inadvertently omitted this detail from the manuscript. In our study, the probe was fixed to an articulating arm, ensuring perfect stability throughout the imaging process. We have now included this clarification in the revised manuscript (see Lines 143 – 144, 557 – 560). Additionally, Dr. Christopoulos is among the first scientists in the States to perform fUSI in non-human primates. His team used a similar approach, securing the probe to an articulating arm and placing it on the dura for brain imaging. In the present study, the probe was likewise attached to an articulating arm and positioned over the spinal cord for transverse imaging. This setup eliminates motion artifacts and ensures reliable data acquisition. The procedure is also described in our recently published Neuron paper [1].

The issue of independence between training and testing datasets in the SVM analysis remains inadequately addressed despite the proposed change of strategy. The low-frequency nature of bladder filling signals inherently creates strong inter frames temporal correlations, making successive frames dependent and challenging the validity

of the model's generalization. The authors should address this limitation explicitly in the manuscript.

Thank you for the observation and suggestion. The reviewer raises an important point, and we acknowledge that the current experimental design limits our ability to generalize our model to predict future bladder pressure changes, given that we are working with a single low-frequency micturition cycle, which may involve inter-frame dependencies. As mentioned in the first revision, we employed a transductive learning approach, as we only have one micturition cycle. Our focus is on reconstructing bladder pressure within that cycle, without aiming to generalize to other datasets, given the time constraints imposed by the clinical protocol, which prevented the inclusion of additional cycles. To address the reviewer's concern, we have updated the Results section to explicitly highlight these potential challenges (Lines 263 – 270). We have also discussed the limitation of the SVM-r model on the Discussion section (Lines 506 – 514).

The manuscript would benefit from a more comprehensive discussion of all those limitations. The authors defend their methodological choices, but the manuscript itself should transparently acknowledge these challenges. For instance, discussing the potential implications of motion correction, probe stabilization, and strong temporal correlations in the SVM analysis would provide a more balanced and critical evaluation of the study.

We agree with the reviewer's comments. Note that we added the ***limitations and clinical challenges*** section in the previous revision, which we have now enriched with the reviewer's suggestions. As we explained above, the probe was mounted on an articulating surgical arm and positioned over the spinal cord. This approach, which was used in our previous study on human spinal cord imaging [1], is similar to the method employed by Christopoulos (corresponding author) and colleagues from Caltech and Physics for Medicine Paris in non-human primate studies [2]. Additionally, we have addressed concerns regarding motion artifacts in the revision by 1) including a movie of the pD images during the experiment, both before and after pre-processing for motion correction (movie V1, patient P4). We also extracted mean Δ SCBV signals from a bladder pressure-correlated region and an adjacent non-correlated region across the entire duration of the experiment (Fig. S3 C, D; patient P2). Our findings show that the average percentage Δ SCBV curve from the bladder pressure-correlated region demonstrates a gradual increase and decrease during bladder filling and emptying, while the Δ SCBV waveform from the adjacent non-correlated region remains flat throughout the urodynamic experiment. Please see the revision for addressing these limitations on section ***limitations and clinical challenges*** (Lines 498 – 505). Finally, we addressed the limitation with the current urodynamic experiment (i.e., only one

micturition cycle) and the SVM-r analysis to predict future bladder pressure curves (Lines 506 – 514).

Minor Suggestions:

The saturation visible in the correlation maps, even if not indicative of exact $+1/-1$ values, can be visually misleading. Are the data capped to a specific threshold or not? The explanation provided for the use of FDR correction is appreciated but could be more straightforward. Citing previous literature is not a particularly strong methodological argument.

Thank you for your suggestion. We confirm that a correlation threshold was selected to capture about the top 5% of both positive and negative correlations of spinal cord pixels to the bladder pressure (i.e., $r > 0.43$, $z\text{-score} > 0.49$ for positive correlations and $r < -0.46$, $z\text{-score} < -0.53$ for negative correlations). We selected this threshold to avoid overcrowding the activation map and to better visualize areas where the pD signal is strongly correlated with the bladder pressure. This approach is widely used in neuroimaging to enhance visualization and highlight the most relevant activations. Our team [1,2,3], along with other research groups [4], has previously employed similar methods in both brain and spinal cord studies. Additionally, we have revised the manuscript to clarify the FDR correction, explicitly detailing how the Benjamini-Hochberg (BH) procedure was applied to control for false positives while maintaining statistical power (see Lines 200 – 211).

While the authors have made substantial progress in addressing previous concerns, critical issues related to motion artifacts and the SVM analysis require further attention. The inclusion of a broader discussion of limitations would also enhance the manuscript's transparency and credibility. I appreciate the authors thoughtful responses thus far and encourage further revisions to resolve these remaining concerns. Pending these revisions, I would support the manuscript acceptance.

In response to your concerns, we have carefully addressed the critical issues related to motion artifacts and the SVM analysis, ensuring a thorough discussion of these aspects in the manuscript. Additionally, we have expanded the limitations section to enhance the transparency and credibility of our study. We believe that these revisions have further strengthened the manuscript, and we deeply appreciate your time and effort in providing such valuable feedback.

References

1. Agyeman KA, Lee DJ, Russin J, Kreydin EI, Choi W, Abedi A, Lo YT, Cavaleri J, Wu K, Edgerton VR, Liu C, Christopoulos VN. Functional ultrasound imaging of the human spinal cord. *Neuron*. 2024 May 15;112(10):1710-1722.e3.
2. Norman SL, Maresca D, Christopoulos VN, Griggs WS, Demene C, Tanter M, Shapiro MG, Andersen RA. Single-trial decoding of movement intentions using functional ultrasound neuroimaging. *Neuron*. 2021 May 5;109(9):1554-1566
3. Griggs WS, Norman SL, Deffieux T, Segura F, Osmanski BF, Chau G, Christopoulos V, Liu C, Tanter M, Shapiro MG, Andersen RA. Decoding motor plans using a closed-loop ultrasonic brain-machine interface. *Nat Neurosci*. 2024 Jan;27(1):196-207
4. Imbault M, Chauvet D, Gennisson JL, Capelle L, Tanter M. Intraoperative Functional Ultrasound Imaging of Human Brain Activity. *Sci Rep*. 2017 Aug 4;7(1):7304.

REVIEWERS' COMMENTS

Reviewer #2 (Remarks to the Author):

The authors have provided compelling evidence and videos to demonstrate that the observed activation reflects localized spinal cord activity, rather than motion or other artifacts. The correlation threshold was explained, and the methods were better clarified. The limitations and clinical challenges have been significantly expanded.

Video SV1: Maybe have arrows to the most apparent motion artifacts in both videos, and somehow quantify the reduction in the motion fluctuations.

We are grateful for effort of the reviewer to provide remarks and suggestions during the review process that we believe significantly improve our work. We have addressed the final comment by adding arrows to indicate the most apparent motion artifacts in both videos (see Movie 1 and lines 608–609) and by quantifying the reduction in motion fluctuations (see lines 613–616). These changes help clarify the robustness of our motion correction approach and strengthen the overall quality of the manuscript.

Reviewer #3 (Remarks to the Author):

The authors have thoroughly addressed all the concerns raised. I commend their efforts in implementing the requested changes and their important and valuable contribution to the field.

We thank the reviewer for their thoughtful feedback and support throughout the review process. We appreciate their recognition of our effort and are grateful for their support of our work and its contribution to the field.